# Revisiting Photometric Ambiguity for Accurate Gaussian-Splatting Surface Reconstruction

Jiahe Li [1 2]   Jiawei Zhang [1]   Xiao Bai [1]   Jin Zheng [1 3]   Xiaohan Yu [4]   Lin Gu [5]   Gim Hee Lee [2]

## Abstract

Surface reconstruction with differentiable rendering has achieved impressive performance in recent years, yet the pervasive photometric ambiguities have strictly bottlenecked existing approaches. This paper presents AmbiSuR, a framework that explores an intrinsic solution upon Gaussian Splatting for the photometric ambiguity-robust surface 3D reconstruction with high performance. Starting by revisiting the foundation, our investigation uncovers two built-in primitive-wise ambiguities in representation, while revealing an intrinsic potential for ambiguity self-indication in Gaussian Splatting. Stemming from these, a photometric disambiguation is first introduced, constraining ill-posed geometry solution for definite surface formation. Then, we propose an ambiguity indication module that unleashes the self-indication potential to identify and further guide correcting undercon­strained reconstructions. Extensive experiments demonstrate our superior surface reconstructions compared to existing methods across various challenging scenarios, excelling in broad compatibility. *Project:* https://fictionarry.github.io/AmbiSuR-Proj/.

## 1. Introduction

Recovering geometry from multi-view images has been a long-term challenging problem in the fields of machine learning (Hartley & Zisserman, 2003; Zheng et al., 2014; Cheng et al., 2024; Zhou & Ni, 2025; Hu & Han, 2025). With the development of differentiable rendering (Milden-

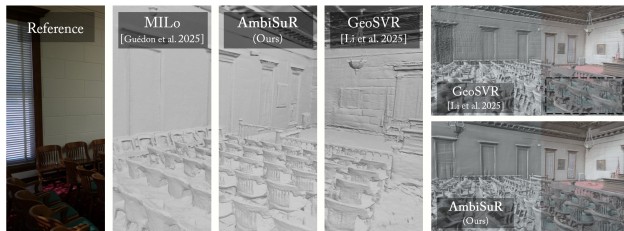

Figure 1. In **challenging** scenarios with ambiguous photometric constraints, previous methods lose the capability to identify the correct surfaces even under priors, leading to a noticeable performance drop with erroneous reconstructions. Instead, AmbiSuR stands out by delivering accurate geometry with delicate details.

hall et al., 2021), optimization-based methods (Yariv et al., 2020; 2021; Wang et al., 2021) appeared to directly learn the surface geometry from 2D images, and achieve impressive performance. In recent years, surface reconstruction evolves significantly with the introduction of 3D Gaussian Splatting (3DGS) (Kerbl et al., 2023), with remarkable advancements (Guédon & Lepetit, 2024; Huang et al., 2024; Zhang et al., 2026; Yu et al., 2024b; Guédon et al., 2025) in efficient and high-precision geometry recovery for diverse scenarios.

The rationale behind these methods stands on photometric consistency. By minimizing the photometric error between renderings and observations, ideally, a unique solution of scene geometry emerges end-to-end as a result of enforcing multi-view photometric consistency. Nevertheless, photometric ambiguities inevitably and unpredictably exist in the real world due to imperfect consistency, such as insufficient, varied, or lost information in distinct cases, which pose long-standing challenges for 3D reconstruction. Thus, multi-view triangulations of these regions become hard and ill-posed to solve via prevailing optimization-based photometric methods upon Gaussian Splatting, as shown in Figure 1. Previous approaches mainly focus on complex ray modeling in limited cases (Yao et al., 2025; Zhang et al., 2025b) or rough external regularizations (Dai et al., 2024; Li et al., 2024b; Chen et al., 2024b; Zhang et al., 2025a), inadequately countercting the root impact of such ambiguities.

In pursuit of the solution, this paper systematically revisits the photometric ambiguity in Gaussian Splatting surface reconstruction from two distinct foundational perspectives. Through representational and supervisory analyses, our in-

[1] School of Computer Science and Engineering, State Key Laboratory of Complex Critical & Software Environment, Jiangxi Research Institute, Beihang University [2] School of Computing, National University of Singapore [3] State Key Laboratory of Virtual Reality Technology and Systems, Beijing [4] Macquarie University [5] Tohoku University. Correspondence to: Xiao Bai <baixiao@buaa.edu.cn>, Jin Zheng <jinzheng@buaa.edu.cn>.

*Proceedings of the 43rd International Conference on Machine Learning*, Seoul, South Korea. PMLR 306, 2026. Copyright 2026 by the author(s).

vestigation uncovers two built-in representational ambiguities that are exacerbated by imperfect supervision, while simultaneously revealing Gaussian Splatting's intrinsic potential for ambiguity self-indication. Based on these findings of foundational intrinsics, we present **AmbiSuR**, a photometric ambiguity-robust framework that delivers accurate, detailed, and efficient surface reconstruction across various scenarios, while exhibiting superior compatibility.

Forming geometry from images, the ambiguity inherent in the photometric representation acts as a fundamental problem. Investigating the optimization process, we discover two primitive-wise ambiguities that directly manifest as geometry flaws under imperfect supervision. First, a basic ambiguity lies in the Gaussian primitive itself, which retains large and low-opacity tail that may cause adverse overlapping, with weak gradients for correcting. Then, acting highly free and unstructured, the less constrained primitive blending process leaves individuals under-determined, resulting in over-reconstruction instead of recovering the definite surface for hard photometric cases. To resolve these problems, we introduce a principled *Gaussian Splatting Photometric Disambiguation* module that diminishes the overblown primitive by a straightforward truncated projection, and enforces optical property local consistency as in physics, founding the basis for correct geometric surface formation.

Then, we turn to the reconstructed ambiguities captured from supervision, exploring their self-indication capability for compensation of the ambiguous constraints. As a natural inner component, we reveal and suggest taking the free-lunch Spherical Harmonics (SH) as an efficient and effective indicator, leveraging the high-degree coefficients to identify high-risk regions in ambiguity. Based on this, we propose a *Spherical Harmonics Ambiguity Indication* module that unleashes the self-indication capability to support a precise regularizer for the fix of problematic geometry. First, we identify two indicative situations of the SH indicator, which are highly related to the primitives being affected by problematic constraints. With this knowledge, a parameter-wise regularizer is designed for amorphously distributed target primitives, keeping compatibility with various geometry priors for hinting. Isolating ambiguities to protect the remainings, this strategy provides strong compensation for directing reconstructions with unclear constraints.

In summary, our main contributions are as follows:

1. A photometric disambiguation, to address two intrinsic ambiguities in Gaussian Splatting, constraining ill-posed geometry solution for definite surface formation.

2. An ambiguity self-indication, that unleashes the indication potential from SH to identify unclear constraints and compensate reconstructions with geometry priors.

3. An AmbiSuR framework, combining the above to ach-

ieve superior surface reconstruction compared to the state-of-the-arts across various challenging scenarios.

**Conflict of Interest Disclosure:**  None.

## 2. Related Work

**Optimization-Based Surface Reconstruction.** Recovering surfaces from multi-view images has been an important long-standing problem in machine learning. Traditional methods (Hartley & Zisserman, 2003; Schönberger et al., 2016; Jiang et al., 2023; Holalkere et al., 2025) work in a modular pipeline with multiple discrete stages. With the development of differentiable rendering, early optimization-based methods (Park et al., 2019; Yariv et al., 2020; Oechsle et al., 2021; Wang et al., 2021) implicitly learn geometry from images by MLPs upon signed distance functions (SDF). Despite improvements (Li et al., 2023; Wu et al., 2022; Yu et al., 2022; Fu et al., 2022; Wang et al., 2022), weaknesses still lie in low efficiency and network capacity. Then, advanced representations (Müller et al., 2022; Kerbl et al., 2023) have remarkably reshaped progress in the field.

**Surface Reconstruction with Gaussian Splatting.** With the rise of 3D Gaussian Splatting (Kerbl et al., 2023), recent optimization-based researches incorporate this unstructured explicit representation for complex geometry. Early methods (Guédon & Lepetit, 2024; Huang et al., 2024; Dai et al., 2024) focus on enhancing mesh quality by primitive-surface alignment. Integrating 3DGS with SDF (Yu et al., 2024a; Lyu et al., 2024; Xu et al., 2024; Zhang et al., 2024b) gets a better combination of efficiency and surface quality, but encounters over-smoothness and network capacity problems as well. In representation, GOF (Yu et al., 2024b) enables detailed mesh extraction in an opacity field, and MILo (Guédon et al., 2025) simultaneously optimizes a differentiable mesh for better surface suitability. However, their ideal performance depends heavily on high photometric consistency.

**Photometric Ambiguity in Surface Reconstruction.** As photometric constraints can seldom be kept perfectly across multiple views, ambiguities extensively exist and lead to ill-posed geometry reconstructions in the real world, which cause long-standing challenges in textureless regions, reflections, insufficient coverage, darks, and so on. A basic idea (Darmon et al., 2022; Fu et al., 2022; Chen et al., 2024a; Li et al., 2025) is to enlarge consistency on larger local plane by explicit multi-view geometry (Hartley & Zisserman, 2003), but works only where photometrics can still locally match well. Recently, some attempts (Yu et al., 2022; Chen et al., 2024b; Zhang et al., 2025a; Li et al., 2025; Zhang et al., 2025b) notice and introduce powerful geometry foundation models (Eftekhar et al., 2021; Yang et al., 2024; Wen et al., 2025) for regularization, while it's unsolved in 3DGS to identify ambiguous regions to maximize the effect (Li et al.,

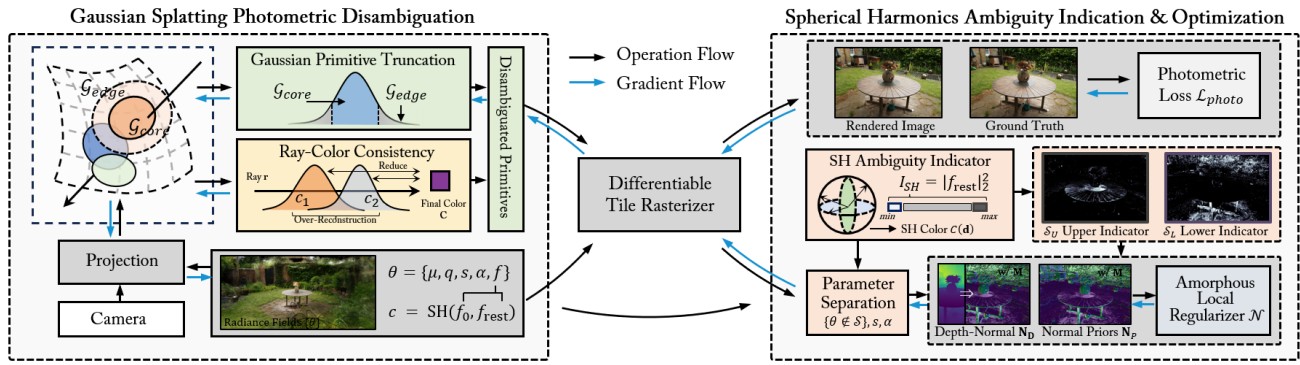

*Figure 2.* **Overview of AmbiSuR**. Our approach stems and operates from two perspectives: (a) *Representationally*, two disambiguation techniques are applied to resolve primitive-wise ambiguity problems in photometric learning, diminishing the overblown Gaussian edges and enforcing optical property local consistency to ensure correct geometry formation. (b) *For ambiguous photometric supervisions*, we reveal and propose taking the free-lunch Spherical Harmonics as effective self-indicators for captured ambiguities, identifying high-risk regions to enable the priors-compatible fine-grained amorphous regularizer for precise geometry compensating and adjustment.

2025). In representation, some methods with ray tracing (Yao et al., 2025; Zhang et al., 2025b) target reflective surfaces, but are designed reflection-oriented only. This paper revisits and reveals the criticality of photometric ambiguity from foundations in Gaussian Splatting, achieving high-quality surface recovery with broad ambiguity robustness.

## 3. Method

### 3.1. Preliminaries

**Gaussian-Splatting Representation.** 3D Gaussian splitting (3DGS) (Kerbl et al., 2023) represents a 3D radiance field with a set of Gaussian primitives. Typically, an $i$-th Gaussian is parameterized as $\theta_i = \{\mu_i, s_i, q_i, \alpha_i, f_i\}$, including a center $\mu$, scaling $s$, rotation $q$, base opacity $\alpha$, and the coefficients $f$ of Spherical Harmonics for per-primitive color $c$. The basis function of the $i$-th primitive $\mathcal{G}_i$ is formulated:

$$\mathcal{G}_i(\mathbf{x}) = e^{-\frac{1}{2}(\mathbf{x}-\mu_\mathbf{i})^T \Sigma_i^{-1}(\mathbf{x}-\mu_\mathbf{i})}, \quad (1)$$

where the covariance matrix $\Sigma$ is calculated from $s$ and $q$.

During rendering, Gaussian Splatting utilizes a point-based rendering to compute the color $\mathbf{C}$ of pixel $\mathbf{x}_p$ by blending $N$ ordered primitives, conducted in a rasterizer:

$$\mathbf{C}(\mathbf{x}_p) = \sum_{i \in N} c_i \tilde{\alpha}_i \prod_{j=1}^{i-1}(1 - \tilde{\alpha}_j); \quad \tilde{\alpha}_i = \alpha_i \mathcal{G}_i^{2D}(\mathbf{x}_p). \quad (2)$$

The rendering opacity $\tilde{\alpha}$ of $N$ ordered intersected primitives to the ray $\mathbf{r}$ across $\mathbf{x}_p$ is calculated by the base opacity $\alpha$, and their projected 2D Gaussians $\mathcal{G}^{2D}$ on image plane.

Geometrically, based on the $\alpha$-blending, pixel depth $\mathbf{D}$ can be derived with per primitive distance. On the shortest axis, the normal of each Gaussian can be defined (Cheng et al., 2024), and thus pixel normal $\mathbf{N}$ can be blended similarly.

**Challenges and Prerequisites.** Classic photometric multi-view consistency assumes the photometric consistency in multi-views, sufficiently distinguishing to locate a unique 3D position in the space. Nevertheless, ambiguous photometrics inevitably and unpredictably exist in the real world, and brings long-standing challenges for 3D reconstruction.

Recently, knowledge from geometry foundation models (Yang et al., 2024; Lin et al., 2025; Wang et al., 2025) delivers external constraints to hint at the structure, which has become an essential tool for the recent high-performance surface reconstructions (Li et al., 2025; Zhang et al., 2025a; Wolf et al., 2024; Chen et al., 2024b; Zhang et al., 2025b). In this work, we inherit the strong but imperfect depth priors as basic prerequisites to investigate the photometric ambiguity problem in the task of high-accuracy surface reconstruction.

### 3.2. Gaussian Splatting Photometric Disambiguation

Learning accurate geometry from photometric supervisions is an important problem for Gaussian Splatting, for which a key is the representational ambiguity. Previous solutions either focus on ray tracing in limited environments (Yao et al., 2025; Zhang et al., 2025b) or geometrically surface-aligned primitives (Huang et al., 2024; Yu et al., 2024b; Guédon et al., 2025). Nevertheless, two photometric-related ambiguity problems still lie in the prevailing pipelines.

**1) Primitive Edge Ambiguity.** First, an inherent ambiguity is in the Gaussian primitive itself, which retains a large area with low-opacity tail at its edge that may cause negative overlapping. Specifically, considering a projected Gaussian primitive $\mathcal{G}^{2D}$ into a core region $\mathcal{G}_{core}$ that contributes primarily, and the edge region $\mathcal{G}_{edge}(\mathbf{x}) \ll 1$ at far position, the pixel-wise opacity $\tilde{\alpha}$ can be represented by two parts:

$$\tilde{\alpha}(\mathbf{x}) = \alpha \mathcal{G}_{core}(\mathbf{x}) \cdot \mathbb{1}_{core} + \alpha \mathcal{G}_{edge}(\mathbf{x}) \cdot \mathbb{1}_{edge} \quad (3)$$

where $\mathbb{1}_{core}$ and $\mathbb{1}_{edge} = 1 - \mathbb{1}_{core}$ denote the binary indicator functions partitioning the spatial support. The

edge regions, only able to obtain a low opacity for unclear representation, may lead to problematic reconstruction.

Projected into different pixels, there's a biased optimization between the $\mathcal{G}_{core}$ and $\mathcal{G}_{edge}$, where the cores are majorly optimized to inflate for regions requiring higher opacity, but causing the large edges to be overblown, as analyzed in Appendix A. This is exacerbated if edge regions are weakly constrained with ambiguity, and pollutes the reconstruction.

**Gaussian Primitive Truncation.** In this work, we utilize a simple yet effective Gaussian Primitive Truncation as solution. It truncates the Gaussians to exclude the edge $\mathcal{G}_{edge}$ from representation, which are less contributing but ambiguous in large area, for the surface reconstruction task.

Specifically, this truncation only retains the core regions $\mathcal{G}_{core}$ during calculating the rendering opacity $\tilde{\alpha}_{\mathcal{T}}$ to act the $\tilde{\alpha}$ in Eq. (2), while the edges are dropped:

$$\tilde{\alpha}_{\mathcal{T}}(\mathbf{x}) = \alpha \mathcal{G}_{core}(\mathbf{x}) \cdot \not\Vdash_{core}. \qquad (4)$$

To ensure the stable and consistent optimization, the binary indicator $\not\Vdash_{core}$ is set for each $i$-th Gaussian, determined by its statistical confidence boundary of standard deviation $\sigma$:

$$\not\Vdash_{core} = \not\Vdash(\|\mathbf{x} - \mu_i\| \leq \gamma \sigma_i), \qquad (5)$$

where $\gamma$ (= 2 in this work) determines the distance. Hence, the primitive ambiguity on the edges is relieved in a concise architecture-agnostic way, thereby resisting the overblown.

**2) Photometric Blending Ambiguity.** The second problem regards an ill-posed primitive-wise color learning under 3DGS's pixel-wise photometric loss. Solely supervising projected primitives from blended color may work fine for view synthesis, but become ambiguous for the definite surface.

Specifically, in previous 3DGS-based approaches, the per-view inverse problem $\min_{\{\theta\}} \|\mathbf{C} - \mathbf{C}_{gt}\|$ with ground-truth $\mathbf{C}_{gt}$ to blended color $\mathbf{C}$ is actually ill-posed for surface reconstruction. Such pixel photometric loss is satisfied as long as the weighted sum matches $\mathbf{C}_{gt}$. As in Figure 2, free color blending imposes a constraint only on the aggregated result, leaving individual primitives under-determined.

This ambiguity facilitates erroneous shortcuts by over-reconstructed and ill-posed geometries of redundant primitives, simulating view-dependent effects via complex occlusion relationships rather than recovering the definite surface.

**Ray-Color Consistency.** Ideally, primitives representing a valid local surface should exhibit similar optical properties to the surface's, which could reject the accumulation of incorrect blending. Stemming from this, we introduce a Ray-Color Consistency that poses a primitive-wise constraint during photometric learning to diminish this ambiguity.

Given the ray $\mathbf{r}$ and the intersected set of $N$ Gaussians $\Theta = \{\theta_i\}_1^N$, the $\alpha$-blending process can be interpreted as

the statistical accumulation along the ray. The blending weight $w_i$ for the $i$-th Gaussian serves as the probability mass function of the ray terminating at that primitive. Consequently, the rendered pixel color $\mathbf{C}$ in Eq. (2) represents the expected value $\mathbb{E}$ of the emitted colors $c$ along the ray:

$$\mathbf{C} = \mathbb{E}[c] = \sum_{i=1}^{N} w_i c_i; \; w_i = \tilde{\alpha}_i \prod_{j=1}^{i-1}(1 - \tilde{\alpha}_j). \quad (6)$$

Ray-Color Consistency constrains the divergence of the per-view color distribution along the ray, by the weighted variance of the emitted colors. To ensure effectiveness, all variables are detached from gradient calculation besides $c_i$.

$$\mathcal{R}(\mathbf{r}) = \mathbb{E}\left[\|c - \mathbb{E}[c]\|_2^2\right] = \sum_{i=1}^{N} w_i \|c_i - \mathbf{C}\|_2^2. \quad (7)$$

This single-view constraint $\mathcal{R}$ enforces the intersected surface of the ray is blended by primitives with similar photometric properties regarding view-dependent color, reducing blending ambiguity during optimization surface forming.

### 3.3. Spherical Harmonics Ambiguity Indication

Despite representational disambiguation, misleading supervisions are still unsolved. Geometry priors help, yet existing incorporation will shrink the effect and introduce errors for 3DGS. In this section, we explore a universal and generalizable methodology for ambiguity indication from image capturing and precise regularization for geometry fixing.

**Learnable SHs are Photometric Ambiguity Indicator.** In most variants of 3DGS, the color of each Gaussian is represented as a function of the viewing direction $\mathbf{d}$ using spherical harmonics (SH). In this work, we suggest it as a *free-lunch* photometric ambiguity indicator.

Specifically, spherical harmonics form an orthogonal basis for functions on the sphere, where the zeroth-order component $l = 0$ corresponds to the isotropic component, i.e., the view-independent color, and higher-order components capture deviations from this ideal perfect sphere assumption. Accordingly, we decompose the color function $C$ as:

$$C(\mathbf{d}) = \bar{C} + C_{\text{rest}}(\mathbf{d}), \quad C_{\text{rest}}(\mathbf{d}) = \sum_i \beta_i Y_i(\mathbf{d}), \quad (8)$$

where $\bar{C}$ denotes the view-independent mean color, and $Y_i$ represents spherical harmonic basis functions of degree $l \geq 1$ with index $i$. The corresponding coefficients $\beta_i$ collected in $f_{\text{rest}}$ encode view-dependent color components.

To quantify how drastically the color deviates from a perfectly view-consistent appearance, we measure the squared color inconsistency integrated over the viewing sphere:

$$\int_{S^2} |C(\mathbf{d}) - \bar{C}|^2 d\omega = \sum_i \beta_i^2 \int_{S^2} Y_i(\mathbf{d})^2 d\omega \propto \sum_i \beta_i^2. \quad (9)$$

As derived in Appendix B, given the orthogonality, this quantity is proportional to the squared L2 norm of the higher-degree SH coefficients, up to a basis-dependent constant.

Here $d\omega$ denotes the area element on the unit sphere $S^2$.

Derived from this, we suggest a direct indicator $I_{SH}$ of photometric ambiguity based on high-degree SH coefficients:

$$I_{SH} = \sum_i \beta_i^2 = \|f_{\text{rest}}\|_2^2, \quad (10)$$

where distinct values statistically indicate different photometric situations captured by the corresponding Gaussian primitives, serving for later selective priors regularization.

**Dual-End Indication.** Then, $I_{SH}$ is leveraged to identify high-ambiguity primitives from two indicative situations.

*1) Upper Indicator.* From the derivation, primitives with a high value in $I_{SH}$ directly refer to suffering from inconsistent constraints, either ambiguity in supervision or caused by inaccurate reconstruction, as in Figure 3. We define this set of high-risk primitives $\mathcal{S}_U^{(t)}$ at iteration $t$. As the coefficients evolve dynamically, the distribution of $I_{SH}$ varies rapidly. For robustness, we employ a dynamic selection by a small set of the top $\eta_U$-percentile of $I_{SH}$ statistics, iteratively:

$$\mathcal{S}_U = \left\{ k \mid I_{SH}^{(k)} > \mathcal{P}\left(1 - \eta_U; \mathbf{I}_{SH}\right) \right\}, \quad (11)$$

where $k$ denotes index, $\mathbf{I}_{SH}$ represents the collection of indicator values from all primitives at step $t$, and $\mathcal{P}(q; \mathbf{I})$ denotes the value at the $q$-th percentile of the set $\mathbf{I}$.

*2) Lower Indicator.* Conversely, primitives with extremely low $I_{SH}$ may relate to problematic regions as well. During optimization process, a compromise state would be intermediately achieved that partially records photometric residual by the view-dependent SH, even for Lambertian surfaces (Appendix C). This makes $I_{SH}$ indicator counterintuitively not close to $0$ when well-constrained, while an extremely low $I_{SH}$ often correlates to inadequate optimization with few photometric supervision or incorrect appearances baked, which helps us to indicate the uncertain regions.

Similarly, we identify this set of potentially problematic primitives $\mathcal{S}_L^{(t)}$ by targeting the small bottom $\eta_L$-percentile:

$$\mathcal{S}_L = \left\{ k \mid I_{SH}^{(k)} < \mathcal{P}\left(\eta_L; \mathbf{I}_{SH}\right) \right\}. \quad (12)$$

Then, indications from these dual ends compose primitives $\mathcal{S} = \mathcal{S}_U \cup \mathcal{S}_L$ that require external prior compensation. Practically, only a small set of primitives is affected, which keeps the regularization efficient and the overall quality robust to the instability that the shifting $I_{SH}$ may have.

**Amorphous Local Regularizer.** Applying $I_{SH}$, our goal is to precisely regularize the amorphously distributed primitive targets, compatible with multiple types of geometry priors. To this end, the regularizer is built for depth-derived normal (Huang et al., 2024), whose normal prior can be easily derived from various prevailing priors (e.g., monocular/multi-view depth, stereo matching, normal estimation, etc.), with

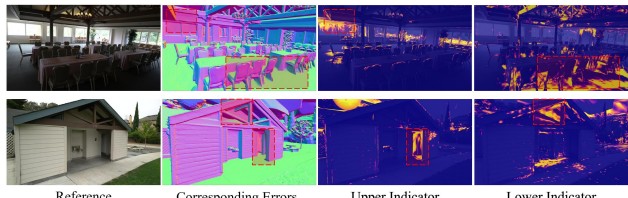

Reference    Corresponding Errors    Upper Indicator    Lower Indicator

*Figure 3.* **Illustration of Dual-End Indication.** On the free-lunch $I_{SH}$, ambiguous regions are identified by dual sets of primitives, accordingly indicating risky reconstructions probably with errors.

minimal locality to enable precise control.

*Parameter Separation.* For precise regularization, we first apply an explicit fine-grained parametric separation per primitive. Specifically, since only ambiguious primitives in $\{\theta_i \mid i \in \mathcal{S}\}$ are targeted, we freeze the parameters of the remainings $\{\theta_i \mid i \notin \mathcal{S}\}$ to avoid negative effect. Then, to enforce regularization precisely on the core surface-related attributes, parameters of opacity $\alpha$ and scaling $s$ are excluded, ensuring stable improvements with sharper details.

*Amorphous Mask.* Then, regularization is designed on a normal loss, limited to the corresponding amorphous regions. Specifically, we first binarily mark the primitives in $\mathcal{S}$ to obtain the distribution map projected at the view:

$$\mathbf{M} = \sum_{i=1}^N \mathbb{K}(i \in \mathcal{S}) \, \tilde{\alpha}_i \prod_{j=1}^{i-1} (1 - \tilde{\alpha}_j). \quad (13)$$

Accommodating the amorphously distributed primitives, it is applied softly to identify the fragmented and discrete targets in the normal loss map, as an amorphous mask:

$$\mathcal{N} = \text{Mean}(\mathbf{M} \cdot (1 - \mathbf{N_D} \cdot \mathbf{N}_P)), \quad (14)$$

where $\mathbf{N_D}$ is the normal map derived from the rendered depth map $\mathbf{D}$, and $\mathbf{N}_P$ denotes the normal map priors extracted from foundation geometry models.

After that, building upon the SH ambiguity indicator, the Amorphous Local Regularizer $\mathcal{N}$ is gained to locally regularize high photometric-ambiguous primitives that may be amorphously distributed with high precision control.

### 3.4. Optimization Objectives

The overall optimization objectives are composed of basic backbone photometric loss $\mathcal{L}_{photo}$ with the geometric regularization $\mathcal{L}_{geo}$ upon geometry priors, the sum of proposed Ray-Color Constraint $\mathcal{R}$ from each ray in Eq. (7), and the Amorphous Local Regularizer $\mathcal{N}$ from Eq. (14):

$$\mathcal{L} = \mathcal{L}_{photo} + \tau \mathcal{L}_{geo} + \mu_1 \mathcal{N} + \mu_2 \mathcal{R} \quad (15)$$

In this work, we uniformly set the weights of $\tau = 0.1$, $\mu_1 = 0.1$, and $\mu_2 = 1e - 5$, respectively.

**Model Variants.** Upon the superior applicability, we propose a standard *AmbiSuR* model with metric depth in $\mathcal{L}_{geo}$

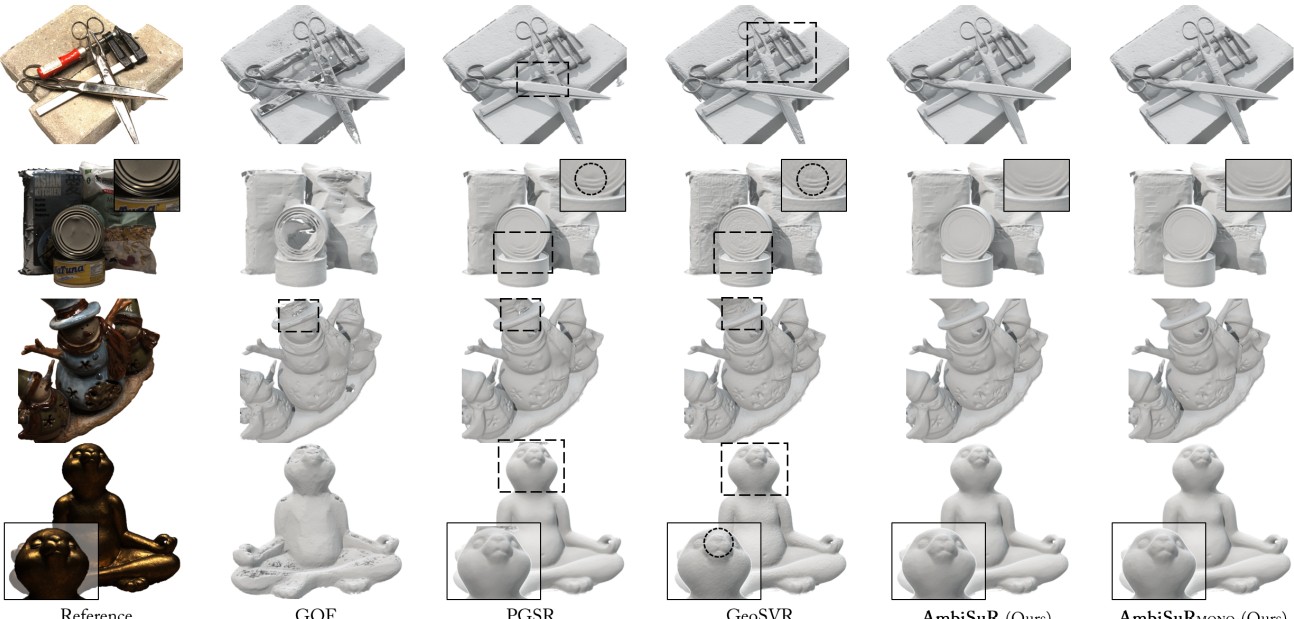

*Figure 4.* **Reconstructed Mesh Comparison on the DTU** (Jensen et al., 2014) **Dataset**. AmbiSuRs consistently reconstruct high-quality smooth surfaces with accurate details, especially in ambiguous cases where even strong prior-enhanced baselines like GeoSVR degrade.

*Table 1.* **Quantitative Comparison on the DTU** (Jensen et al., 2014) **Dataset**. Outperforming the implicit, voxel-based, and Gaussian Splatting baselines, AmbiSuRs exhibit robust reconstruction with the highest quality on the Chamfer distance. Best results are highlighted.

| | Method | 24 | 37 | 40 | 55 | 63 | 65 | 69 | 83 | 97 | 105 | 106 | 110 | 114 | 118 | 122 | Mean | Time |
|---|---|---|---|---|---|---|---|---|---|---|---|---|---|---|---|---|---|---|
| Implicit | NeuS (Wang et al., 2021) | 1.00 | 1.37 | 0.93 | 0.43 | 1.10 | 0.65 | 0.57 | 1.48 | 1.09 | 0.83 | 0.52 | 1.20 | 0.35 | 0.49 | 0.54 | 0.84 | >12h |
| | Neuralangelo (Li et al., 2023) | 0.37 | 0.72 | 0.35 | 0.35 | 0.87 | 0.54 | 0.53 | 1.29 | 0.97 | 0.73 | 0.47 | 0.74 | 0.32 | 0.41 | 0.43 | 0.61 | >128h |
| | Geo-NeuS (Fu et al., 2022) | 0.38 | 0.54 | 0.34 | 0.36 | 0.80 | 0.45 | 0.41 | 1.03 | 0.84 | 0.55 | 0.46 | 0.47 | 0.29 | 0.36 | 0.35 | 0.51 | >12h |
| | MonoSDF (Yu et al., 2022) | 0.66 | 0.88 | 0.43 | 0.40 | 0.87 | 0.78 | 0.81 | 1.23 | 1.18 | 0.66 | 0.66 | 0.96 | 0.41 | 0.57 | 0.51 | 0.73 | 6h |
| Explicit | 2DGS (Huang et al., 2024) | 0.48 | 0.91 | 0.39 | 0.39 | 1.01 | 0.83 | 0.81 | 1.36 | 1.27 | 0.76 | 0.70 | 1.40 | 0.40 | 0.76 | 0.52 | 0.80 | 0.2h |
| | GOF (Yu et al., 2024b) | 0.50 | 0.82 | 0.37 | 0.37 | 1.12 | 0.74 | 0.73 | 1.18 | 1.29 | 0.68 | 0.77 | 0.90 | 0.42 | 0.66 | 0.49 | 0.74 | 1h |
| | GS2Mesh (Wolf et al., 2024) | 0.59 | 0.79 | 0.70 | 0.38 | 0.78 | 1.00 | 0.69 | 1.25 | 0.96 | 0.59 | 0.50 | 0.68 | 0.37 | 0.50 | 0.46 | 0.68 | 0.3h |
| | VCR-GauS (Chen et al., 2024b) | 0.55 | 0.91 | 0.40 | 0.43 | 0.97 | 0.95 | 0.84 | 1.39 | 1.30 | 0.90 | 0.76 | 0.92 | 0.44 | 0.75 | 0.54 | 0.80 | ~1h |
| | PGSR (Chen et al., 2024a) | 0.36 | 0.57 | 0.38 | 0.33 | 0.78 | 0.58 | 0.50 | 1.08 | 0.63 | 0.59 | 0.46 | 0.54 | 0.30 | 0.38 | 0.34 | 0.52 | 0.5h |
| | MILo (Guédon et al., 2025) | 0.43 | 0.74 | 0.34 | 0.37 | 0.80 | 0.74 | 0.70 | 1.21 | 1.22 | 0.66 | 0.62 | 0.80 | 0.37 | 0.76 | 0.48 | 0.68 | 0.4h |
| | GeoSVR (Li et al., 2025) | 0.32 | 0.51 | 0.30 | 0.33 | 0.71 | 0.48 | 0.42 | 1.03 | 0.62 | 0.56 | 0.33 | 0.46 | 0.30 | 0.34 | 0.32 | 0.47 | 0.8h |
| | **AmbiSuR$_{MONO}$ (Ours)** | 0.32 | 0.48 | 0.31 | 0.33 | 0.65 | 0.48 | 0.45 | 0.98 | 0.61 | 0.53 | 0.36 | 0.44 | 0.29 | 0.35 | 0.34 | 0.46 | 0.6h |
| | **AmbiSuR (Ours)** | 0.32 | 0.48 | 0.31 | 0.33 | 0.65 | 0.48 | 0.42 | 0.97 | 0.61 | 0.54 | 0.36 | 0.44 | 0.29 | 0.34 | 0.34 | 0.46 | 0.6h |

for the strongest performance upper-bound, and a variant *AmbiSuR-Mono* that incorporates the easily obtained and robust monocular depth with broad compatibility.

Specifically, in AmbiSuR, we use the L1 loss $\mathcal{L}_{geo} = |\mathbf{D} - \mathbf{D}_P|_1$ normalized by scene range with priors $\mathbf{D}_P$, and initialize with its derived point cloud. For AmbiSuR-Mono, we take the patch-depth loss (Li et al., 2026; 2025) for $\mathcal{L}_{geo}$ with monocular depth, under standard SfM initialization.

## 4. Experiments

**Implementation Details.** With high architectural agnosticism, we evaluate our method on the general baseline PGSR (Chen et al., 2024a). Aligned with 3DGS default, each model is trained with 30,000 iterations. Multi-view depths are gained from Depth Anything 3 (Lin et al., 2025). For

the Mono variant, we use the same Depth Anything V2 (Yang et al., 2024) for depth cues, as previous works (Li et al., 2025; Guédon et al., 2025). Defaultly, $\eta_U$ of the upper indicator is set to 5%. $\eta_L$ adopts 10% in DTU for its darker lighting, and 5% for other datasets. TSDF is used for mesh extraction. All experiments are done on RTX 3090 Ti GPUs.

**Datasets.** Adopting standard settings, we use the DTU (Jensen et al., 2014), Tanks and Temples (TnT) (Knapitsch et al., 2017), and Mip-NeRF 360 (Barron et al., 2022) datasets in experiments. Evaluation scene splits keep the same as the standard setting (Yariv et al., 2021; Wang et al., 2021; Li et al., 2023; Huang et al., 2024). Images of DTU and TnT, which are separately preprocessed by 2DGS (Huang et al., 2024) and Neuralangelo (Li et al., 2023). Keeping aligned with previous works, 2× image downsampling is adopted for DTU and TnT, 2× or 4× for indoor and

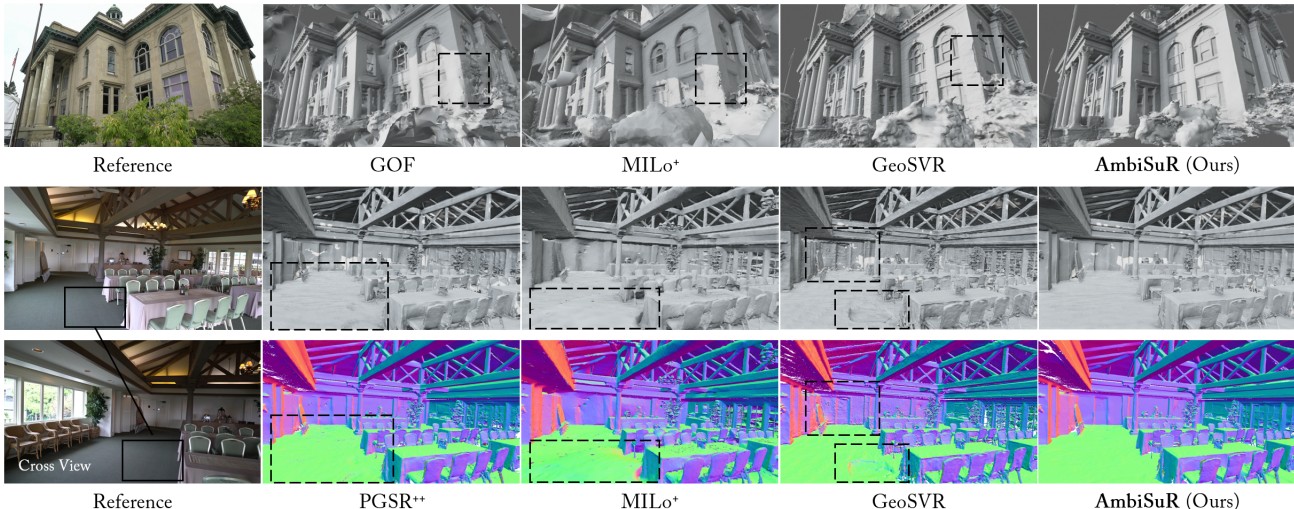

*Figure 5.* **Reconstructed Mesh Comparison on the Tanks and Temples** (Knapitsch et al., 2017) **Dataset** with high-performing baselines. AmbiSuR delivers accurate reconstruction in large-scale scenes that consist of pervasive photometric ambiguities, with outstanding prior exploitation for geometry guidance. MILo⁺ and PGSR⁺⁺ denote boosted by monocular and metric depth priors, respectively.

*Table 2.* **Quantitative Comparison on the Tanks and Temples** (Knapitsch et al., 2017) **Dataset**. Bests for implicit and explicit methods are separately marked. AmbiSuRs achieve the best in F1 score for the superior reconstruction of real-world complex scenarios.

| | SDF | | | Voxel | | Gaussian Splatting | | | | | |
|---|---|---|---|---|---|---|---|---|---|---|---|
| | MonoSDF | Neuralangelo | NeuRodin | SVRaster | GeoSVR | GOF | VCR-GauS | MILo⁺ | PGSR | **AmbiSuR_MONO** | **AmbiSuR** |
| Time | 6h | >128h | 18h | 11m | 68m | 24m | 53m | 131m | 45m | 50m | 49m |
| Barn | 0.49 | 0.70 | 0.70 | 0.35 | 0.68 | 0.51 | 0.62 | 0.61 | 0.66 | 0.67 | 0.67 |
| Caterpillar | 0.31 | 0.36 | 0.36 | 0.33 | 0.49 | 0.41 | 0.26 | 0.40 | 0.44 | 0.51 | 0.52 |
| Courthouse | 0.12 | 0.28 | 0.21 | 0.29 | 0.34 | 0.28 | 0.19 | 0.31 | 0.20 | 0.36 | 0.39 |
| Ignatius | 0.78 | 0.89 | 0.87 | 0.69 | 0.83 | 0.68 | 0.61 | 0.76 | 0.81 | 0.83 | 0.83 |
| Meetingroom | 0.23 | 0.32 | 0.43 | 0.19 | 0.37 | 0.28 | 0.19 | 0.27 | 0.33 | 0.39 | 0.45 |
| Truck | 0.42 | 0.48 | 0.47 | 0.54 | 0.66 | 0.59 | 0.52 | 0.59 | 0.66 | 0.68 | 0.68 |
| *Mean* | 0.39 | 0.50 | 0.51 | 0.40 | 0.56 | 0.46 | 0.40 | 0.49 | 0.52 | 0.57₆ | 0.58₉ |

outdoor scenes in the Mip-NeRF 360 dataset.

### 4.1. Evaluation

**Surface Reconstruction.** To evaluate the performance on surface reconstruction, we conduct experiments on the DTU and TnT datasets, with the metrics of Chamfer distance and F1-score in Table 1 and 2. On the DTU dataset, both AmbiSuR and AmbiSuR-Mono achieve the best overall accuracy beyond all the baselines, including the voxel-based SOTA method GeoSVR (Li et al., 2025) that leverages the same monocular priors as AmbiSuR-Mono. In Figure 4, while the 3DGS-based GOF (Yu et al., 2024b) and PGSR (Chen et al., 2024a) wrongly reconstruct the photometrically ambiguous regions, especially specular, AmbiSuR successfully achieves superior performance under two kinds of priors. Adopting the same monocular depth, GeoSVR performs worse in facing these challenges compared to our quality, and leads to a coarser surface due to the granularity of structured voxel representation. In contrast, our method achieves the best combination of details and accuracy.

On the TnT dataset, AmbiSuRs achieve the bests as well, standing out by delivering comprehensively strong performance. Either with scale-ambiguous monocular or less accurate metric depth, AmbiSuR consistently reconstructs with high accuracy, surpassing current SOTA methods that adopt the same priors (e.g., MILo (Guédon et al., 2025) and GeoSVR (Li et al., 2025)), as shown in Table 2 and Figure 5. Considering the inevitable imperfection of the priors, the shown advantages and robustness verify the value of the proposed photometric ambiguity techniques in this work, which further demonstrate the necessity of our investigation.

**Appearance Reconstruction.** In addition, to evaluate the novel view synthesis capability, we test our method on the Mip-NeRF 360 dataset in Table 3. While surface reconstruction largely improved, our method performs well in rendering. Although appearance fitting may be limited due to the simple ray modeling in 3DGS, our method still achieves a competitive overall quality among all baselines. In Figure 6, the qualitative results show our advantages in the challenging regions that are hardly addressed in previous.

*Table 3.* **Quantitative Results on Mip-NeRF 360 Dataset.** Bests among the surface reconstruction methods are marked with colors.

| | | Outdoor Scenes | | | Indoor Scenes | | |
|---|---|---|---|---|---|---|---|
| | | PSNR ↑ | SSIM ↑ | LPIPS ↓ | PSNR ↑ | SSIM ↑ | LPIPS ↓ |
| NVS | NeRF | 21.46 | 0.458 | 0.515 | 26.84 | 0.790 | 0.370 |
| | Instant NGP | 22.90 | 0.566 | 0.371 | 29.15 | 0.880 | 0.216 |
| | Mip-NeRF 360 | 24.47 | 0.691 | 0.283 | 31.72 | 0.917 | 0.180 |
| | 3DGS | 24.67 | 0.728 | 0.240 | 30.96 | 0.924 | 0.187 |
| | SVRaster | 24.68 | 0.738 | 0.206 | 30.65 | 0.927 | 0.161 |
| Surface Recon. | SuGaR | 22.93 | 0.629 | 0.356 | 29.43 | 0.906 | 0.225 |
| | 2DGS | 24.34 | 0.717 | 0.246 | 30.40 | 0.916 | 0.195 |
| | GOF | 24.82 | 0.750 | 0.202 | 30.79 | 0.924 | 0.184 |
| | VCR-GauS | 24.31 | 0.707 | 0.280 | 30.53 | 0.921 | 0.184 |
| | PGSR | 24.76 | 0.752 | 0.203 | 30.36 | 0.934 | 0.147 |
| | GeoSVR | 24.83 | 0.738 | 0.218 | 30.46 | 0.921 | 0.172 |
| | **AmbiSuR** | 24.79 | 0.752 | 0.202 | 30.06 | 0.928 | 0.159 |

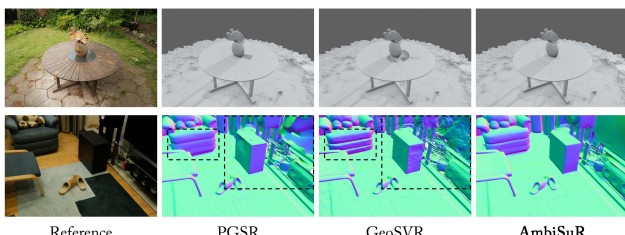

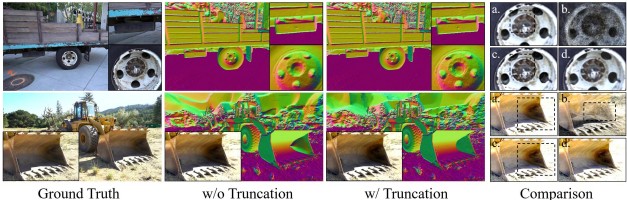

Reference     PGSR     GeoSVR     AmbiSuR

*Figure 6.* **Mesh Comparison on the Mip-NeRF 360 Dataset.** AmbiSuR stands out by well reconstructing in difficult regions.

Ground Truth     w/o Truncation     w/ Truncation     Comparison

*Figure 7.* **Visualized Effect of Gaussian Primitive Truncation.** Right column: **a.** w/o Truncation; **b.** expectedly truncated edges in a.; **c.** excluding components b. from a.; **d.** w/ Truncation. Eliminating ambiguity from edges, this surprisingly direct strategy effectively relieves the ambiguous primitive's overblown problem.

## 4.2. Ablation Study

To verify the effect of the proposed components, we conduct ablation studies with various settings in this section. In Table 4, we show a comprehensive study of each component on the TnT dataset, and Table 5 for a detailed investigation of the SH Ambiguity Indicator on the DTU dataset.

**Gaussian Primitive Truncation.** As a concise and simple strategy, Table 4 shows a surprising improvement in surface accuracy (Item B to C) from it. In Figure 7, we visualize the benefits in detail, which show that the original Gaussian primitives would construct unclear geometry if not receiving strong photometric supervision, relevant to the ambiguity at the edges. By removing these edge ambiguities even without re-training, a direct improvement is shown. The effect achieves best when the Truncation is fully adopted.

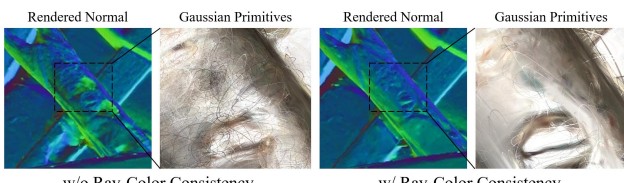

Rendered Normal    Gaussian Primitives    Rendered Normal    Gaussian Primitives

w/o Ray-Color Consistency      w/ Ray-Color Consistency

*Figure 8.* **Visualized Effect of Ray-Color Consistency.** Solely constraining blended color leads to over-reconstruction with erroneous primitives, which can be well controlled by our technique.

*Table 4.* **Ablation Study on the TnT Dataset.** The results demonstrate the effectiveness of AmbiSuR's proposed components.

| Item | Disambi. | | Indicator | | Priors | | Accuracy Metrics | | |
|---|---|---|---|---|---|---|---|---|---|
| | Trunc | RayColor | Naive | SHAmbi | Mono | Multi | Precison ↑ | Recall ↑ | F1-Score ↑ |
| A. | - | - | - | - | - | - | 0.506 | 0.551 | 0.522 |
| B. | - | - | - | - | ✓ | - | 0.530 | 0.575 | 0.547 |
| C. | ✓ | | | | ✓ | | 0.545 | 0.580 | 0.558 |
| D. | ✓ | ✓ | | | ✓ | | 0.552 | 0.589 | 0.566 |
| E. | ✓ | | | ✓ | ✓ | | 0.554 | 0.591 | 0.569 |
| F. | ✓ | ✓ | ✓ | | ✓ | | 0.533 | 0.592 | 0.557 |
| **G.** | ✓ | ✓ | | ✓ | ✓ | | 0.568 | 0.594 | 0.576 |
| **H.** | ✓ | ✓ | | ✓ | | ✓ | 0.579 | 0.608 | 0.589 |

*Table 5.* **Ablation Study of SH Ambiguity Indicator.** The effect and necessity of each included design are quantitatively verified.

| Item | Settings | d-to-s ↓ | s-to-d ↓ | Cf-Dist ↓ |
|---|---|---|---|---|
| A. | Naive | 0.436 | 0.519 | 0.477 |
| B. | G. w/o Dual-End | 0.431 | 0.515 | 0.473 |
| C. | B. + Upper Indicator | 0.423 | 0.516 | 0.469 |
| D. | B. + Lower Indicator | 0.423 | 0.506 | 0.464 |
| E. | G. w/o Amorphous Mask | 0.424 | 0.520 | 0.472 |
| F. | G. w/o Param Sep | 0.423 | 0.517 | 0.470 |
| **G.** | Full Model | 0.419 | 0.504 | 0.461 |

**Ray-Color Consistency.** Table 4 shows an obvious effect of Ray-Color Consistency ("RayColor") in different cases (Item E to G, C to D). For better illustration, Figure 8 visualizes a comparison. In a challenging region where primitives may learned the color poorly, a shortcut geometry may be built to wrongly deal with the case by redundant primitives blended. Our strategy eliminates this ambiguity in photometric learning, reconstructing clear and correct surface.

**SH Ambiguity Indication.** To show the effect of this technique ("SHAmbi"), we compare it to the setting solely adopting the same depth-normal loss uniformly in each view ("Naive"). In Table 4, SHAmbi works effectively to improve the overall quality (Item C to E), even when the accuracy is already at a high level (D to G). Meanwhile, comparing F and G, it's shown that the Naive solution could not achieve the similar effect, and could even harm the already accurate reconstructions, leading to significant performance drop. This demonstrate the effectiveness of our approach. Furthermore, we provide a detailed study on the DTU dataset on Table 5, which verifies the necessity of each component.

# 5. Conclusion

In this work, we present AmbiSuR to explore an intrinsic solution for the photometric ambiguity-robust surface reconstruction. Upon Gaussian Splatting, our work uncovers two built-in primitive-wise ambiguities in representation, and reveals an intrinsic potential for ambiguity self-indication. With these, photometric disambiguation and ambiguity indication modules are introduced, constraining ill-posed geometry solutions, and unleashing self-indication to identify and direct underconstrained regions. In the future, incorporating complex light propagations will be an interesting direction.

# Impact Statement

This paper presents a method for high-quality 3D surface reconstruction from 2D images. So far, we have not discovered direct negative societal impact. However, the increased accuracy of our method brings certain societal considerations. Precise real-world reconstructions may inadvertently capture sensitive personal information, leading to privacy risks. Additionally, the potential for malicious use in generating unauthorized digital copies exists. These societal impacts should be treated with caution. We advocate for the responsible application of this technology.

# Acknowledgements

This work is supported by Jiangxi Provincial Natural Science Foundation 20252BAC250018, Key R&D Program of Jiangxi Province, China (20232BBE50019), and the National Natural Science Foundation of China 62276016, 62372029. Lin Gu is supported by JST Moonshot R&D Grant Number JPMJMS2011 Japan. This research / project is supported by the National Research Foundation, Singapore, under its NRF-Investigatorship Programme (Award ID. NRF-NRFI09-0008), and the Tier 2 grant MOE-T2EP20124-0015 from the Singapore Ministry of Education.

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

# Appendix

## A. Analysis of Primitive Edge Ambiguity

In this section, we analyze the optimization dynamics of a Gaussian primitive to explain the geometric bleeding phenomenon. We quantitatively demonstrate that the gradient at the primitive's boundary is effectively suppressed due to the dominance of exponential opacity decay.

**Shape Gradient Formulation.** Considering in the 2D image space, the gradient of the photometric loss $\mathcal{L}$ with respect to the covariance $\tilde{\Sigma}$ of the projected Gaussian is given by:

$$\frac{\partial \mathcal{L}}{\partial \tilde{\Sigma}} = \sum_{\mathbf{x} \in \mathcal{P}} \Psi(\mathbf{x}) \cdot \frac{\partial \tilde{\alpha}(\mathbf{x})}{\partial \tilde{\Sigma}}, \tag{16}$$

where $\Psi(\mathbf{x})$ denotes the backpropagated errors from the renderer. For a clear analysis, let $\Psi(\mathbf{x})$ temporarily be consistent. The shape update is governed by the sensitivity of opacity to covariance changes:

$$\frac{\partial \tilde{\alpha}(\mathbf{x})}{\partial \tilde{\Sigma}} = \frac{1}{2} \tilde{\alpha}(\mathbf{x}) \left( \tilde{\Sigma}^{-1} (\mathbf{x} - \mu)(\mathbf{x} - \mu)^T \tilde{\Sigma}^{-1} \right). \tag{17}$$

**Quantitative Analysis of Gradient Accumulation.** To rigorously analyze the spatial behavior of the gradient, we define the Mahalanobis distance of $r = \sqrt{(\mathbf{x} - \mu)^T \tilde{\Sigma}^{-1}(\mathbf{x} - \mu)}$. The magnitude of the gradient term in Eq. (17) is proportional to the product of a quadratic distance and the Gaussian exponential. We define the response function as $g(r)$:

$$\|\frac{\partial \tilde{\alpha}}{\partial \tilde{\Sigma}}\|_F = \frac{1}{2}\alpha \cdot e^{-r^2/2} \cdot \|\tilde{\Sigma}^{-1}(\mathbf{x} - \mu)\|_2^2 \quad \propto \quad g(r) = r^2 \cdot e^{-r^2/2}. \tag{18}$$

For anisotropic covariance matrices, this proportionality holds up to a constant factor that depends on the orientation of $\tilde{\Sigma}$, but the functional form $g(r)$ captures the essential decay behavior of the gradient magnitude with respect to $r$.

Analyzing the properties of $g(r)$ in the context of spatial summation, we must account for the linear growth of the integration area. Conceptually, we identify two distinct optimization regimes based on the behavior of $J(r)$:

$$J(r) = r \cdot g(r) = r^3 \cdot e^{-r^2/2}. \tag{19}$$

Let the Mahalanobis bound distance be $r_b$, corresponding to the $\gamma\sigma$ in the main paper. The two conceptual situations are:

*1) Core Region:* This region ($0 \leq r \leq r_b$) refers to the center part of the primitive. The accumulated gradient in the core is:

$$S_{core}(r_b) = \int_0^{r_b} J(r)dr = 2 - e^{-r_b^2/2}(r_b^2 + 2). \tag{20}$$

Along with $r$, this term increases rapidly, receiving strong gradients from errors to fit the photometric content.

*2) Edge Region:* The edge region ($r > r_b$) corresponds to the edges extending into the background. Despite the integration area growing to infinity, the exponential decay dominates. Therefore, the total gradient feedback is strictly bounded:

$$S_{edge}(r_b) = \int_{r_b}^{\infty} J(r)dr = e^{-r_b^2/2}(r_b^2 + 2). \tag{21}$$

As $r_b$ increases, $S_{edge}$ decays exponentially. This indicates that the total correction capacity of the edge is capped and rapidly vanishes for large boundaries.

**Optimization Bias for Geometric Over-expansion.** To determine the inherent bias of the optimization, we analyze the balance between the driving gradient in the core and the corrective ones from the edge. We define the $\eta(r_b)$ as the ratio of the accumulated gradients in the two regions:

$$\eta(r_b) = \frac{S_{core}(r_b)}{S_{edge}(r_b)} = \frac{2 - e^{-r_b^2/2}(r_b^2 + 2)}{e^{-r_b^2/2}(r_b^2 + 2)} = \frac{2e^{r_b^2/2}}{r_b^2 + 2} - 1. \tag{22}$$

The optimization reaches a critical state when the accumulated gradient in the core matches the maximum possible feedback from the tail, with $\eta(r_{crit}) = 1$. This leads to the equation:

$$e^{r_{crit}^2/2} = r_{crit}^2 + 2 \quad \Rightarrow \quad r_{crit} \approx 1.83. \tag{23}$$

Note that $\eta$ is monotonically increasing. When the primitive boundary $r_b$ exceeds the critical distance $r_{crit}$, a structural bias occurs, where the total gradient in the core strictly exceeds the corrective capacity of the edge. Consequently, we obtain the inequality:

$$\Big| \sum_{\mathbf{x} \in \mathcal{G}_{\text{core}}} \nabla_{\tilde{\Sigma}} \mathcal{L} \Big| > \text{ or } \gg \Big| \sum_{\mathbf{x} \in \mathcal{G}_{\text{edge}}} \nabla_{\tilde{\Sigma}} \mathcal{L} \Big| \quad s.t. \ \ r_b > r_{crit}. \tag{24}$$

This trend creates an irreversible bias that minimizing photometric error in the core generates a strong expansion force. Even if the edge region covers a significantly larger spatial area on the screen, its aggregated gradient magnitude is insufficient to correct the expansion. Here we list the typical values to intuitively show this bound from the rapidly vanishing gradient in Table 6 (for simplicity, std $= \sigma$. $\mu = 0$).

Overall, the dominating exponential decay of $S_{edge} : S_{core} = 1/\eta$ indicates that the core strictly overwhelms the edge feedback. Considerable pixels only provide minor corrective feedback even if they are incorrectly covered by the Gaussian's large edge. Therefore, while minimizing photometric error in the core region could lead to expanding, the adjustment of erroneously covered pixels by edges is suppressed heavily.

This imbalance leads to undesired inflation of the large ambiguous edge, as a byproduct of fitting the dominant gradient from the core region. Especially, this could be exacerbated when the edge regions are in weak supervision and lead to suboptimal reconstructions.

*Table 6.* **Relationship List of Two Regions with Different $r_b$.**

| $r_b$ | $S_{edge} : S_{core}$ | $\mathcal{G}(r_b\sigma)$ |
|---|---|---|
| 1.50 | 2.22 : 1 | 0.324 |
| *1.83* | *1.00 : 1* | *0.187* |
| 2.00 | 0.68 : 1 | 0.135 |
| 2.50 | 0.22 : 1 | 0.043 |
| 3.00 | 0.06 : 1 | 0.011 |
| 4.00 | 3e-3 : 1 | 0.000 |

## B. Analysis of High-Degree Spherical Harmonics Indicator

In most variants of 3DGS, the color of each Gaussian is represented as a function of the viewing direction $\mathbf{d}$ using spherical harmonics (SH). In this work, we suggest it as a *free-lunch* photometric ambiguity indicator.

Specifically, spherical harmonics form an orthogonal basis for functions on the sphere, where the zeroth-order component $l = 0$ corresponds to the isotropic component, i.e., the view-independent color, and higher-order components capture deviations from this ideal perfect sphere assumption. Accordingly, we decompose the color function $C$ as:

$$C(\mathbf{d}) = \bar{C} + C_{\text{rest}}(\mathbf{d}), \quad C_{\text{rest}}(\mathbf{d}) = \sum_i \beta_i Y_i(\mathbf{d}), \tag{25}$$

where $\bar{C}$ denotes the view-independent mean color, and $Y_i$ represents spherical harmonic basis functions of degree $l \geq 1$ with index $i$. The corresponding coefficients $\beta_i$ collected in $f_{\text{rest}}$ encode view-dependent color components.

To quantify how drastically the color deviates from a perfectly view-consistent appearance, we measure the squared color inconsistency integrated over the viewing sphere:

$$E_{\text{c}} = \int_{S^2} |C(\mathbf{d}) - \bar{C}|^2 d\omega = \int_{S^2} \Big( \sum_i \beta_i Y_i(\mathbf{d}) \Big) \Big( \sum_j \beta_j Y_j(\mathbf{d}) \Big) d\omega. \tag{26}$$

Here $d\omega$ denotes the infinitesimal area element on the unit sphere $S^2$. By exploiting the linearity of the integral and rearranging the summation order, we expand the expression into:

$$E_{\text{c}} = \sum_i \sum_j \beta_i \beta_j \int_{S^2} Y_i(\mathbf{d}) Y_j(\mathbf{d}) d\omega. \tag{27}$$

Due to the orthogonality of the spherical harmonics basis, the inner product $\int_{S^2} Y_i Y_j d\omega$ vanishes for distinct indices $i \neq j$. By introducing the normalization constant $\lambda_i = \int_{S^2} Y_i(\mathbf{d})^2 d\omega$ for each basis function, the formulation collapses into a single sum:

$$E_{\text{c}} = \sum_i \sum_j \beta_i \beta_j (\lambda_i \delta_{ij}) = \sum_i \beta_i^2 \lambda_i. \tag{28}$$

$\lambda_i$ is a structural constant determined solely by the basis definition. Considering the common case with normalization such that $\lambda_i$ is constant across the band, the integrated squared inconsistency is directly proportional to the squared Euclidean

norm of the coefficient vector. Especially, in 3DGS, the spherical harmonics are usually implemented using an orthonormal basis, which implies $\lambda_i = 1$ for all components. Without loss of generality, the relationship can be formulated:

$$E_{\text{c}} \propto I_{SH} = \sum_i \beta_i^2 = \|f_{\text{rest}}\|_2^2. \tag{29}$$

Derived from this, we suggest a direct indicator $I_{SH}$ of photometric ambiguity based on high-degree SH coefficients, where distinct values statistically indicate different photometric situations captured by the corresponding Gaussian primitives.

From a spectral perspective, this relationship can also be derived from Parseval's theorem on the sphere, which states that the total energy of the function is conserved in the spectral domain. Explained from this end, the indicator $I_{SH}$ corresponds to the total energy of the view-dependent color variations distributed across the entire viewing sphere, building a solid relationship between the indicator and the photometric behavior of the primitive.

## C. Analysis of Low $I_{SH}$ in Optimization

In this section, we provide a formal analysis of the compromise solution encountered during Gaussian Splatting optimization. We specifically illustrate how the capacity constraints inherent in progressive densification lead to geometric errors being baked into view-dependent appearance parameters.

**Errors from Structural Under-Reconstruction.** Gaussian Splatting optimizes scene representation via a strategy of progressive densification. Consider an intermediate optimization step $t$ where the set of Gaussian primitives $\Theta = \{\theta\}$ has size $N_t$. Let $N^*$ denote the theoretical number of primitives required for a faithful geometric reconstruction of the scene with perfect appearance. In the under-reconstructed state that covers most of the optimization process, we have $N_t \ll N^*$.

Due to this sparsity, the geometry-consistent, view-independent parameters $\Theta_{\text{base}} = \{\theta - \{f_{\text{rest}}\}\}$ possess limited representational capacity. For this part, an irreducible photometric error when rendering the scene is denoted as $\mathcal{E}_{struct}$. And therefore, except for the truly view-dependent appearances (we mark as $\mathcal{E}_{v.d}$), this error exists even considering the ideal case with a view-independent Lambertian assumption, where only requires $C(\mathbf{d}) = \bar{C}$ for the ground-truth reconstruction in Eq. (8):

$$\mathcal{E}_{struct} = \mathbf{P}_{gt} - \phi(\mathbf{0}; \Theta_{\text{base}}) - \mathcal{E}_{v.d}, \tag{30}$$

where $\phi(\mathbf{d}; \Theta)$, receiving view direction $\mathbf{d}$ and parameters $\Theta$, denotes all-image rendering function from Eq. (2) and (8) by $c = C(\mathbf{d})$, while here $\mathbf{d} = \mathbf{0}$ as $f_{\text{rest}}$ is excluded, and therefore view-dependent parts are unavailable. As above, $\|\mathcal{E}_{struct}\| > 0$ because the sparse geometry $\Theta$ fails to represent all the spatial information in the ground truth images $\mathbf{P}_{gt}$.

**Compromise Optimization for View-Dependent Parameters.** In practice, the optimization objective minimizes the total photometric loss $\mathcal{L}$ with respect to both geometric parameters $\Theta_{\text{base}}$ and view-dependent appearance parameters $\Theta_{\text{rest}} = \{f_{\text{rest}}\}$. The rendering output $\mathbf{P}_{pred}$ can be conceptually decomposed into a base geometric term and a view-dependent modulation term, lifting an view-dependent part isolatedly related to $\Theta_{\text{rest}}$:

$$\mathbf{P}_{pred} = \phi(\mathbf{0}; \Theta_{\text{base}}) + \Delta_{\text{rest}}(\mathbf{d}; \Theta) \quad \Rightarrow \quad \Delta_{\text{rest}}(\mathbf{d}; \Theta) = \phi(\mathbf{d}; \Theta_{\text{base}} + \Theta_{\text{rest}}) - \phi(\mathbf{0}; \Theta_{\text{base}}) \tag{31}$$

To minimize the target loss $\|I_{pred} - I_{gt}\|$, the optimizer exploits the degrees of freedom in $\Theta_{\text{rest}}$ to bridge the photometric gap caused by the structural under-reconstruction. Considering situation $\Theta_{\text{base}}$ has already been optimized to be locally optimal, this residual still establishes to force an fitting of the irreducible error $\mathcal{E}_{struct}$ by $\Delta_{\text{rest}}(\mathbf{d}; \Theta)$,:

$$\min_{\Theta_{\text{base}}, \Theta_{\text{rest}}} \|I_{gt} - I_{pred}\| \quad \Rightarrow \quad \min_{\Theta_{\text{rest}}} \|\mathcal{E}_{struct} + \mathcal{E}_{v.d} - \Delta_{\text{rest}}(\mathbf{d}; \Theta)\|. \tag{32}$$

Since the non-zero error $\mathcal{E}_{struct}$ can not be solved by the view-independent parameters in $\Theta_{\text{base}}$, the optimizer will only force the $\Theta_{\text{rest}}$ to overfit it, according to the misleading dynamic signals from the varying view direction $\mathbf{d}$.

As perfect reconstructions are not available at initialization, once the optimization starts, this phenomenon will happen in the regions being supervised by the photometrics. As a result, considering the inevitable light change and material reflections ($\mathcal{E}_{v.d}$) in the real world additionally, an extremely low $I_{SH}$ often correlates with regions receiving insufficient photometric supervision or ill-posed geometry where appearances are incorrectly baked, which helps us to indicate the uncertain regions.

# D. Additional Ablation Study

## D.1. Depth Priors

Compatible with diverse priors, especially the depth with the proposed two variants, we conducted an ablation study including prevailing depth foundation models to verify the performance under different providing priors. In the ablation study, monocular depths Depth Pro (Bochkovskii et al., 2024) and Depth Anything V2 (Yang et al., 2024) are applied to AmbiSuR-Mono ("Mono" in Type), and metric depths from multi-view models VGGT (Wang et al., 2025), MVSAnywhere (Izquierdo et al., 2025) and Depth Anything 3 (Lin et al., 2025) are applied to the standard AmbiSuR ("Metric" in Type). Table 7 reports the results on DTU dataset. For additional comparison, we listed the cases with the applied depth priors enhancement

*Table 7.* **Ablation Study on Depth Prior Model.** [1]As a universal geometry model, VGGT does not receive known poses and provides less robust dense depth, yet our method extracts benefits from it.

| Type | Depth Model | D-to-S $\downarrow$ | S-to-D $\downarrow$ | Chamfer $\downarrow$ |
|---|---|---|---|---|
| Base | Mono (DAV2) | 0.434 | 0.519 | 0.477 |
| Base | Metric (DA3) | 0.440 | 0.520 | 0.480 |
| Mono | DepthPro | 0.412 | 0.506 | 0.459 |
| Mono | DepthAnythingV2 | 0.419 | 0.504 | 0.461 |
| Metric[1] | VGGT | 0.430 | 0.505 | 0.468 |
| Metric | MVSAnywhere | 0.422 | 0.505 | 0.463 |
| Metric | DepthAnything3 | 0.419 | 0.504 | 0.461 |

for our adjusted baseline model ("Base" in Type), without any proposed components, to emphasize our contributions.

The results show that our method exhibits high robustness in the kinds of priors, across different models, types, and qualities. Compared to the baseline with high-quality Depth Anything V2 (DAV2) and Depth Anything 3 (DA3) models, our method consistently achieves obviously superior performance in different cases, with constant robustness. Especially, our study applies the VGGT model as one situation, which is famous at its universal capability of high-quality feedforward pose estimation and point clouds, but obtains much worse dense depth quality than other depth-oriented models. Even with this prior, our method still performs well and achieves better results (0.468) than previous SOTA methods. Especially, this is partially contributed by the locality of Amorphous Local Regularizer: 1) Erasing the reliance on the metric information to avoid the prior errors in the difficult metric estimation; 2) strictly isolating the ambiguous primitives in parameter-wise, from the effective indicator $I_{SH}$, for the precise regularization applying.

## D.2. Gaussian Primitive Truncation

As derived in Appendix A, the bound of $\gamma\sigma$ in Sec. 3.2 (corresponding to $r_b$) splits one Gaussian primitive into two conceptual regions with different dynamics with a lower bound, of which the exact value for $\gamma$ in usage is the target we further studied here. In Table 8, we report the performance under different $\gamma$ on the TnT dataset, along with the corresponding value of $\mathcal{G}$ at the bound.

*Table 8.* **Ablation Study on the Bound in Gaussian Primitive Truncation.** Value of $\mathcal{G}$ at the corresponding bound is reported.

| Bound $\gamma$ ($\sigma$) | $\mathcal{G}$ | Precision $\uparrow$ | Recall $\uparrow$ | F1-Score $\uparrow$ |
|---|---|---|---|---|
| 1.5 | 0.32 | 0.564 | 0.591 | 0.574 |
| 2.0 | 0.14 | 0.579 | 0.608 | 0.589 |
| 2.2 | 0.09 | 0.578 | 0.608 | 0.588 |
| 2.5 | 0.04 | 0.576 | 0.605 | 0.586 |
| 3.0 | 0.01 | 0.572 | 0.599 | 0.580 |
| 4.0 | 3e-4 | 0.567 | 0.597 | 0.577 |
| $+\infty$ | 0.00 | 0.566 | 0.596 | 0.576 |

The results show a consistent conclusion with our theoretical analysis in Appendix A, where the cases behave differently at two sides of the theoretically critical distance about $r_{crit} \approx 1.83$. Compared to the other situations, including the one without truncation ($\gamma = +\infty$), the performance has a significant drop at $\gamma = 1.5 < r_{crit}$, where the representational capability of the core region has been harmed either by receiving insufficient gradient. Instead, the method at the derived desired range of $r_{crit} < \gamma = 2.0, 2.2, 2.5$ robustly performs well even with a large change in $\mathcal{G}$ from 0.04 to 0.14 (3.5×). Then, by the increase of $\gamma$, the ratio of truncated edge region shrinks quickly, being close to the case without truncation, which leads to the performance falling back to the basic. Additionally, more qualitative comparisons between w/ or w/o truncation are shown in Figure 9. Generally, the overall trend well matches the theoretical analysis of the biased optimization as in Appendix A and Sec. 3.2, which provide the solid foundation for the shown robustness and effectiveness of the truncation with bound of $\gamma\sigma$ in our method.

## D.3. Spherical Harmonics Ambiguity Indication

In the additional ablation study of the Spherical Harmonics Ambiguity Indication for Sec. B, we focus on the impact of the dual-end indication. In our method, we propose taking two small sets of the primitives in the top $\eta_U$-percentile and bottom

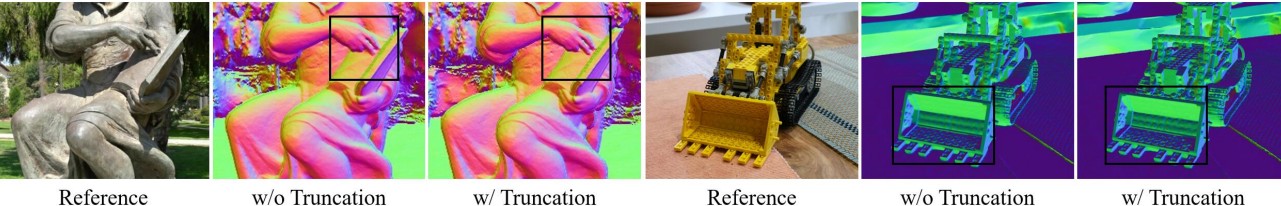

| Reference | w/o Truncation | w/ Truncation | Reference | w/o Truncation | w/ Truncation |

*Figure 9.* **Additional Qualitative Comparison between w/ or w/o Gaussian Primitive Truncation.** Our technique improves reconstructing clear details and more accurate surfaces that are hard to be resolved by previous methods, without extra costs.

$\eta_L$-percentile iteratively to be included in the regularizer. Here we provide a further study about the size of the two taken ends to verify the robustness of this technique.

The results from experiments on DTU dataset are reported in Table 9. To help better comparison, the naive solution of adopting the depth-normal loss uniformly in per view without the dual-end indication in the amorphous local regularizer, the same as the setting in the main paper, is also reported in the table. From the results, it is shown that the dual-end indication shows high robustness to different settings of $\eta_U$ and $\eta_L$, constantly providing significant improvements in the

*Table 9.* **Ablation Study on the Dual-End Size of $\eta_U$ and $\eta_L$.** Constant qualities demonstrate high robustness of our design.

| Upper 1-$\eta_U$ | Lower $\eta_L$ | D-to-S ↓ | S-to-D ↓ | Chamfer ↓ |
|---|---|---|---|---|
| Naive | | 0.436 | 0.519 | 0.477 |
| 90% | 10% | 0.422 | 0.504 | 0.463 |
| 98% | 10% | 0.421 | 0.504 | 0.462 |
| 95% | 10% | 0.419 | 0.504 | 0.461 |
| 95% | 5% | 0.420 | 0.506 | 0.462 |
| 95% | 15% | 0.417 | 0.504 | 0.461 |

reconstruction quality. Especially, the effectiveness is proved of only taking two small amounts (5% or even 2%) of primitives into the regularization per iteration, which also demonstrates the high efficiency of our regularization design, along with the precise targeting of the problematic regions. Overall, as shown in the ablation studies, this approach provides a concise and robust methodology for isolating the risky primitives by rapid and small-amount iterations, achieving superior performance in adjusting under-constrained regions while protecting the well-reconstructed regions unaffected.

### D.4. Ray-Color Consistency

To better illustrate the effect of the Ray-Color Consistency, here we show another example to demonstrate its contribution. As analyzed in the main paper, lacking primitive-wise constraint that leads to under-determined photometric attributes, in the case without Ray-Color Consistency in Figure 10, it can be observed that the primitives blended for a reflective surface are not composed with consistent colors, but producing half-transparent dark primitives for occlusion from certain

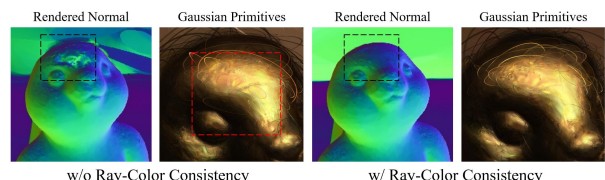

| Rendered Normal | Gaussian Primitives | Rendered Normal | Gaussian Primitives |
| w/o Ray-Color Consistency | | w/ Ray-Color Consistency | |

*Figure 10.* **Additional Visualization of Ray-Color Consistency**, recovering geometry by inhibiting attribute divergence.

perspectives, and light-colored primitives at deep. Although these complex occlusion changes simulate a desired photometric appearance, the geometry is indeed flawed. After solely applying Ray-Color Consistency without any other changes, this problem is effectively relieved along with the improved primitive-wise photometric consistency, as shown in Figure 10.

## E. Experimental Details

### E.1. Implementation Details

In the experiment, we implement our method with PyTorch and CUDA kernels upon the codebase of PGSR (Chen et al., 2024a). Aligned with the 3DGS's default, each model is trained with $30,000$ iterations. During optimization, we use the standard 3DGS photometric loss combined with L1 and D-SSIM for $\mathcal{L}_{photo}$, and start applying $\mathcal{L}_{geo}$ at $1,000$ iterations. We implement the Ray-Color Consistency $\mathcal{R}$ in the CUDA backpropagation for efficiency, working until the stop of densification at $15,000$ iterations to avoid over-regularization. To ensure a moderate reconstruction has been formed for adjustment, Amorphous Local Regularizer $\mathcal{N}$ is adopted after $7,000$ iterations. As in the main paper, the weights of the loss terms are set to $\tau = 0.1$, $\mu_1 = 0.1$, and $\mu_2 = 1e-5$, respectively. In Gaussian Primitive Truncation, we set the support of distance to $2\sigma$. Defaultly, in dual-end indication, $\eta_U$ of the upper indicator is set to 5%. $\eta_L$ is adopted as 10% in DTU that has darker lighting environments, and 5% for other datasets. All models are trained on RTX 3090 Ti GPUs.

For the standard AmbiSuR, we apply metric depths as priors from Depth Anything 3 (Lin et al., 2025), using the weight DA3NESTED-GIANT-LARGE. To handle the long sequences, we apply a sliding window of size 200, and align the scale with the first refernce frame. Point clouds from the depth back-projection with the top 20% confidence are used as initialization. For the Mono variant, we use Depth Anything V2 (Yang et al., 2024) of weight Depth-Anything-V2-Large, the same as in previous baseline works (Li et al., 2025; Guédon et al., 2025), to provide monocular depth cues, following GeoSVR (Li et al., 2025) to use the patch-wise depth loss as a basic geometry constraint. This variant uses the default SfM point cloud initialization, consistent to the other compared 3DGS-based works.

### E.2. Datasets

Our evaluation use the standard settings (Yariv et al., 2021; Wang et al., 2021; Fu et al., 2022; Li et al., 2023; Huang et al., 2024; Yu et al., 2024b; Li et al., 2025) for surface reconstruction on DTU (Jensen et al., 2014), Tanks and Temples (TnT) (Knapitsch et al., 2017), and Mip-NeRF 360 (Barron et al., 2022) datasets. Specifically, 1) 15 scans in DTU are selected for evaluation, with IDs of 24, 37, 40, 55, 63, 65, 69, 83, 97, 105, 106, 110, 114, 118, 122, downsampling all the images to the half-resolution images for training. For preprocessing, we use the widely-used off-the-shelf public data[1] from 2DGS (Huang et al., 2024) in the format of COMLAP (Schönberger et al., 2016; Schönberger & Frahm, 2016). 2) For TnT, 6 discriminative scenes ("Barn", "Caterpillar", "Courthouse", "Ignatius", "Meetingroom", "Truck") are in use for evaluation, with the provided training images, camera poses and ground truth point cloud. Preprocessing is done through the script[2]. 3) All the 9 scenes in Mip-NeRF 360 are used for evaluation, using the official COLMAP-format data. Images are downsampled $2\times$ or $4\times$ following (Kerbl et al., 2023) separately for indoor ("bonsai", "counter", "kitchen", "room") and outdoor ("bicycle", "garden", "flowers", "stump", "treehill") scenes.

**Metrics.** Standardly, Champer distance is used to measure the reconstruction of DTU dataset, and F1-Score for TnT. All measurements are conducted with the widely-used public off-the-shelf toolkits for TnT[3] and DTU[4], the same as the most competitive baselines (Chen et al., 2024a; Li et al., 2025) for fairness. For the novel view synthesis on Mip-NeRF 360, metrics implemented in 3DGS (Kerbl et al., 2023) are used, the same as in most previous explicit methods (Huang et al., 2024; Yu et al., 2024b; Chen et al., 2024a; Li et al., 2025).

### E.3. Baselines

The important baselines in the comparison are as follows: 1) implicit SDF-based methods: NeuS (Wang et al., 2021), Neuralangeo (Li et al., 2023), Geo-NeuS (Fu et al., 2022), MonoSDF (Yu et al., 2022), NeuRodin (Wang et al., 2024); 2) explicit voxel-based methods: SVRaster (Sun et al., 2025), GeoSVR (Li et al., 2025); 3) explicit 3DGS-based methods: 2DGS (Huang et al., 2024), GOF (Yu et al., 2024b), GS2Mesh (Wolf et al., 2024), VCR-GauS (Chen et al., 2024b), PGSR (Chen et al., 2024a)), MILo (RaDe-GS (Zhang et al., 2026) (Base)) (Guédon et al., 2025).

In the comparison, to emphasize our contribution, we additionally evaluates previous methods with geometry priors, marked as MILo⁺ (used in Figure 5, Table 2, and also Figure 1 without explicit mark) and PGSR⁺⁺ (in Figure 5). For MILo⁺, we apply the method using the officially implemented depth order loss with Depth Anything V2. For PGSR⁺⁺, we apply the same $\mathcal{L}_{geo}$ and point cloud initialization as in AmbiSuR from Depth Anything 3.

**Source of Reported Results.** In the part of the baseline methods, we prioritize using the officially provided quantitative and qualitative results, from either official papers or code repositories, to ensure fairness by reducing errors in the reproduction process. Following previous works, on TnT dataset, we report the quantitative results of Geo-NeuS reproduced by Neuralangelo, which was not officially evaluated on this benchmark. For the missing qualitative results of the involved methods, we follow the open-source code instruction to reproduce the corresponding surface models and get the renderings. Training times are from official papers if available. For the remaining, we measure the time in our experimental environment.

## F. Discussion to Voxel Geometric Uncertainty

Focusing on differentiating the degree of prior regularization, the most related technique to our SH Ambiguity Indication in Sec. 3.3 is the Voxel-Uncertainty Depth Constraint in the recent sparse voxel-based GeoSVR (Li et al., 2025), while our approach requires fewer limitations that can be applied in the highly free and unstructured Gaussian Splatting.

---

[1] https://drive.google.com/drive/folders/1SJFgt8qhQomHX55Q4xSvYE2C6-8tFll9
[2] https://github.com/NVlabs/neuralangelo/blob/main/DATA_PROCESSING.md
[3] https://github.com/isl-org/TanksAndTemples/tree/master/python_toolbox/evaluation
[4] https://github.com/hbb1/2d-gaussian-splatting/tree/main/scripts/eval_dtu

Instead of relying on the structural assumption, our Spherical Harmonics Ambiguity Indication directly focuses on the learned photometric ambiguities from ill-posed supervisions, which obtains higher applicability with fewer limitations. This allows our approach to work on the highly free and unstructured Gaussian Splatting methods across different scenarios, regardless of the diverse and complex primitive distributions for spatial representation.

## G. Qualitative Comparison

In the appendix, we provide additional qualitative mesh visualizations in this section. Specifically, in Figure 11 and 12, we show the reconstructed mesh in the Mip-NeRF 360 dataset. Figure 13 and 14 visualize all the reconstructed objects in the DTU dataset. Figure 15 is reported for the TnT dataset. Additionally, we provide more video demos in the supplementary material. We recommend watching it for a better visualization effect.

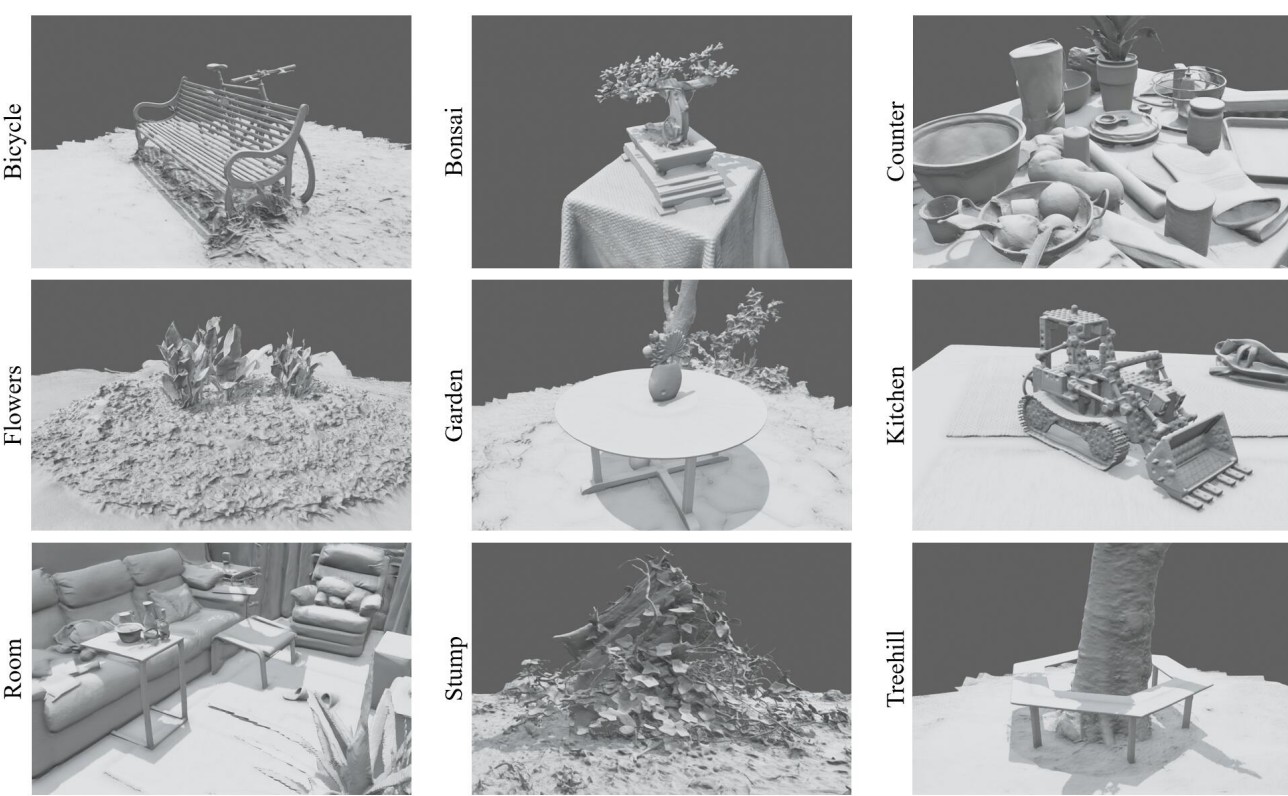

*Figure 11.* **Visualization of the Reconstructed Meshes (without Vertice Color) on the Mip-NeRF 360 (Barron et al., 2022) Dataset.**

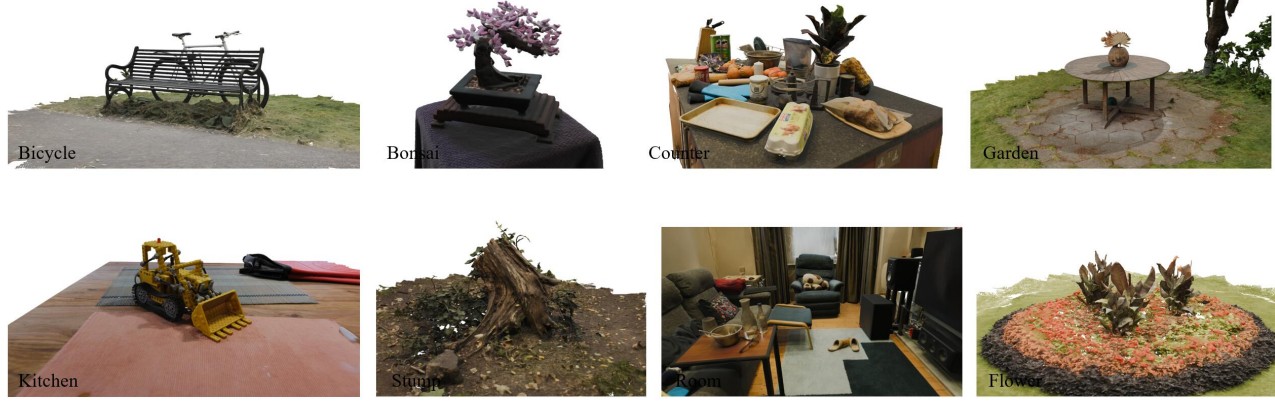

*Figure 12.* **Visualization of the Reconstructed Meshes (with Vertice Color) on the Mip-NeRF 360 (Barron et al., 2022) Dataset.**

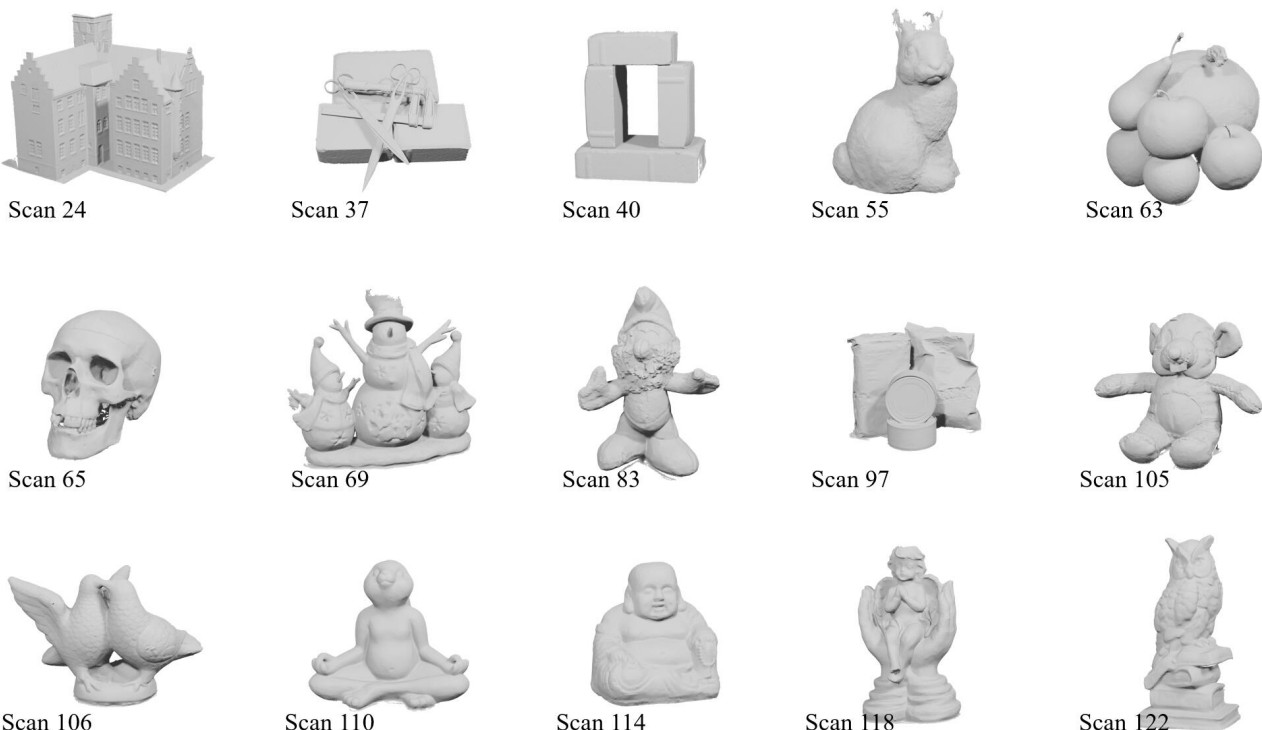

*Figure 13.* **Visualization of the Reconstructed Meshes (without Vertice Color) on the DTU (Jensen et al., 2014) Dataset.**

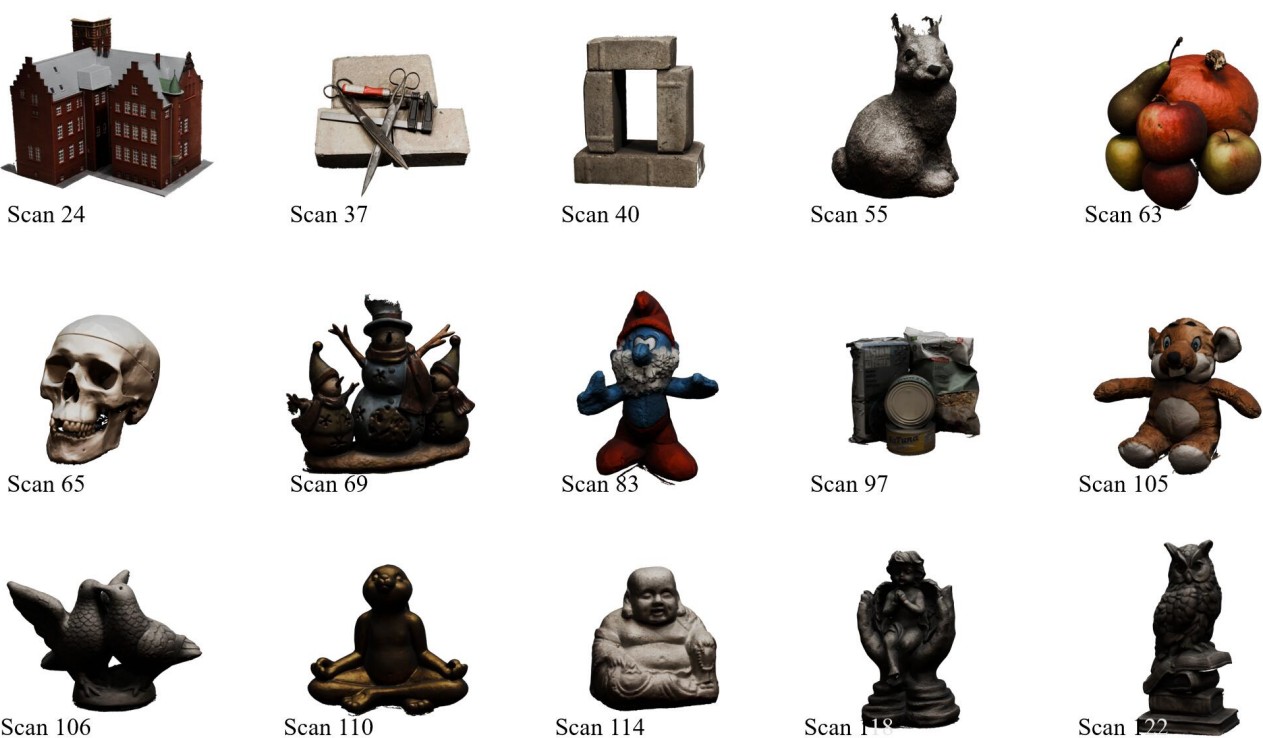

*Figure 14.* **Visualization of the Reconstructed Meshes (with Vertice Color) on the DTU (Jensen et al., 2014) Dataset.**

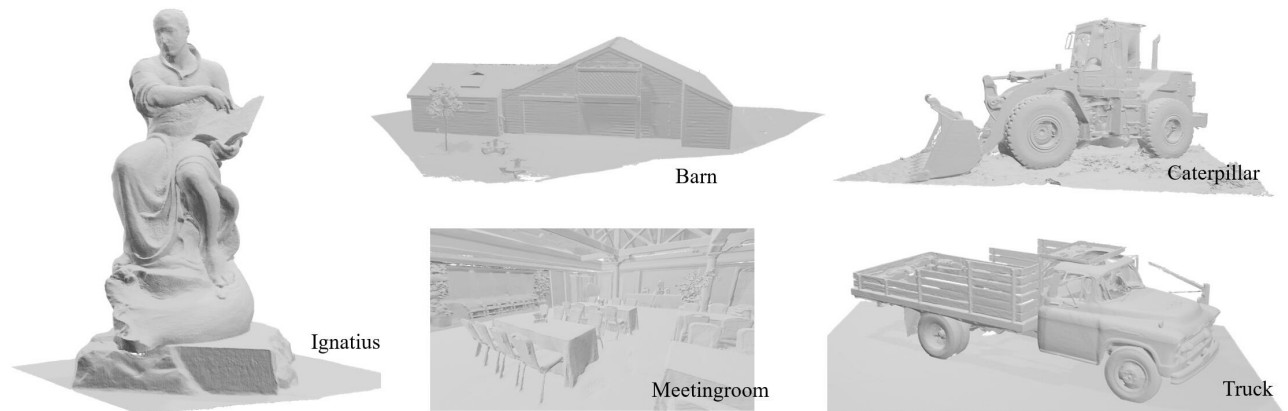

*Figure 15.* **Visualization of the Reconstructed Meshes on the Tanks and Temples Dataset (Knapitsch et al., 2017).**

## H. Effect on Non-Lambertian Surfaces

As we discussed later, nowadays identifying transparent surfaces, especially with heavy refraction, is one of the most difficult problems that can hardly be resolved without explicit multi-bounce light transport calculation and known environment. Facing this hard problem, Ray-Color Consistency actually provides benefits, relieving the incorrect over-reconstruction from overfitting. To show this in-the-wild performance, here we additionally evaluate our method on the representative EnvGS dataset (Xie et al., 2025), which contains real-world transparent surfaces. As shown in Figure 16, Ray-Color Consistency effectively helps reconstruct transparent surfaces with no additional degradation.

Meanwhile, SH Ambiguity Indicator shows strong effect in this case, especially for surface with complex optical properties, significantly improving the geometry in the wild beyond the naive solutions. These show the effectiveness of our proposals.

Nevertheless, we acknowledged this is an opening and challenging issue that our method currently cannot well solve, especially for refractive surfaces, which is a valuable future direction.

## I. Stability of Percentile-Based Primitive Selection

In our test, the percentile-based primitive selection of Eq. (11, 12) keep high stability during the optimization. While primitives vary and SH distribution shifts, our indicator in practice focuses on the ambiguities from supervision, which keep stable over time, as visualized in Figure 17. Here we additionally conduct ablations on the update intervals in Table 10, which show that while selection sets evolve frequently, this dynamic process remains stable and benefits reconstruction accuracy. Combined with the verification in Table 9, the percentile-based primitive selection exhibits high stability in the training process.

*Table 10.* **Effect on Update Interval** for Dual-End Indication.

| Update per Steps | Cf. ↓ | d2s ↓ | s2d ↓ |
|---|---|---|---|
| 1 | **0.461** | 0.419 | **0.504** |
| 10 | **0.461** | **0.417** | 0.505 |
| 30 | 0.462 | 0.418 | 0.506 |
| Only After Densification | 0.466 | 0.428 | 0.504 |
| Vanilla (for Reference) | 0.477 | 0.436 | 0.519 |

## J. Performance of Amorphous Mask in Detecting Ill-Constrained Regions

In the paper, we analyzed the robustness and performance of amorphous mask by ablating the individual component (Table 5.E), percentile (Table 9), and visualized its effects in Figure 2 and 3. To quantify the detected ill-constrained regions that lead to low reconstruction accuracy, we conduct an exploratory study here by measuring the depth error in different regions. However, to expose the inaccuracy and avoid underfitting, we'd like to note that this can only be measured where regularization is not applied and only when the optimization is finished (rather than the desired intermediate). Under such non-ideal cases, the consistently concentrated error detection with low area proportion shows the strong robustness and performance of the amorphous mask.

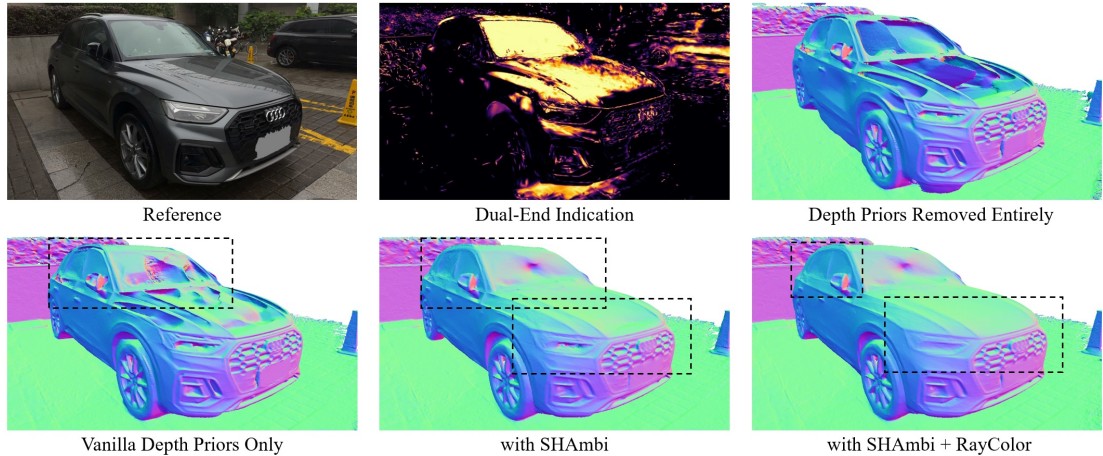

*Figure 16.* **Effects of Different Components in Transparent and Reflective Surfaces.**

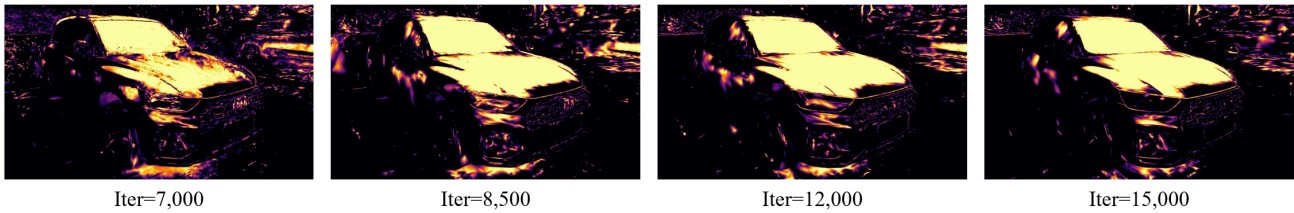

*Figure 17.* **Stability of SH Ambiguity Indication from Applying to the End of Densification.**

*Table 11.* (Exploratory Quantitative Study) **Masked Depth Error versus the Average**. High robustness performance is shown in the non-ideal non-regularization use cases. Measured on the 30,000th iteration without applying $\mathcal{N}$ compared to the aligned GT point cloud.

| Terms | Barn | Caterpillar | Courthouse | Ignatius | Meetingroom | Truck |
|---|---|---|---|---|---|---|
| L1 Error In Mask | **+ 39.28%** | **+ 46.87%** | **+ 14.28%** | **+ 29.63%** | **+ 15.06%** | **+ 96.29%** |
| L1 Error Out of Mask | - 7.14% | - 3.12% | - 0.02% | - 0.00% | - 5.47% | - 3.70% |
| Mask Area Ratio | 13.3% | 9.7% | 20.9% | 3.2% | 27.7% | 6.3% |

## K. Reconstruction Effect for Delicate Structures from Primitive Truncation

Gaussian Primitive Truncation would not degrade the capability of reconstructing delicate and semi-transparent structures. As analyzed in Table 8 in the appendix, this technique exhibits an obvious effect by only needing to conduct truncation where the Gaussian weight has already dropped to a much lower level than the core ($\sim$ 0.1 to 0.01), which ensures the representation capability is maintained as the initials. Secondly, in Gaussian Splatting, the transparency of reconstruction is mainly controlled by the individual opacity property for each entire primitive, therefore the truncation will not affect this part. Here we provide the corresponding qualitative results in Figure 18 and 16, which demonstrate our strong or even better reconstruction regarding the distant foliage, atmospheric effects, and semi-transparent glasses.

## L. Photometric Ambiguity Indication under Sparse Views

The Dual-End Indication in the proposed SH Ambiguity Indicator takes primitives with a large norm of high-degree SH coefficients into consideration, which includes cases of SH overfitting like in sparse-view scenarios, where various recent works focused on (Li et al., 2024a; Zhang et al., 2024a; Gu et al., 2026; Wu et al., 2025). Similar to the shown samples in Figure 3, the upper indicator can effectively reflect the inaccurate geometries that using overfitted SH to mimic the photometric appearance, which also works in the sparse-view cases. In Figure 19, we test the situation using 9 sparse views and compare the indication to the dense-view situation. Besides the observed ambiguity varies due to the changed number of views, the results show that the upper indicator shows high consistency to the situation with dense views. This demonstrates the effectiveness of our method. *Nevertheless, we note that* the definitely correct reconstruction solely from photometric

w/o Truncation · · · · · · w/ Truncation

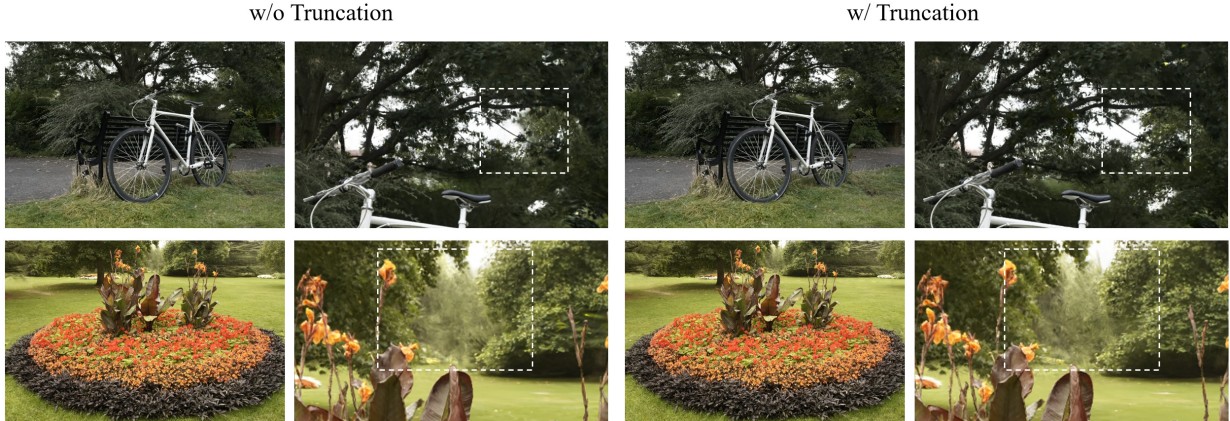

*Figure 18.* **Novel View Synthesis of Gaussian Primitive Truncation for Distant Foliage and Slightly Foggy Atmospheric Effects.**

Reference · · · Dense Views · · · Sparse Views · · · · Reference · · · Dense Views · · · Sparse Views

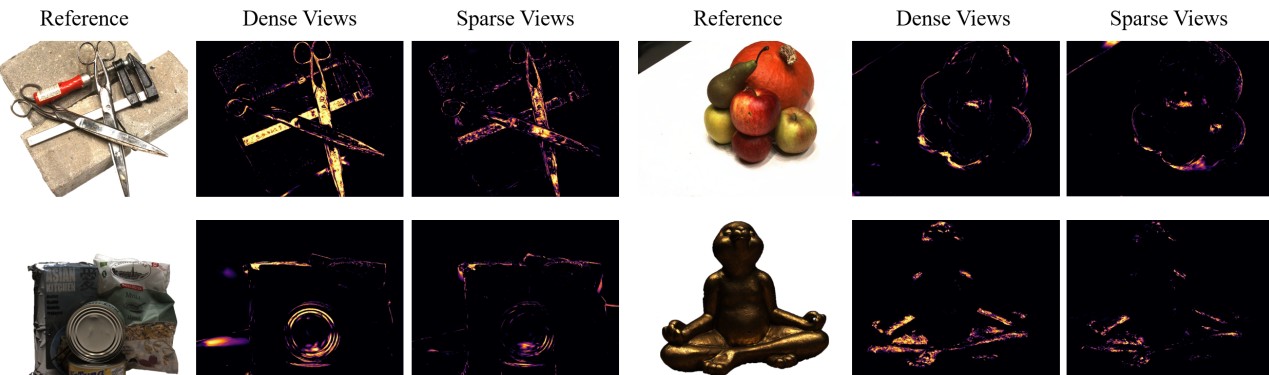

*Figure 19.* **Comparison of Upper Indicator under Sparse-View (9 Views) and Dense-View Settings.**

constraints is much less under sparse views, which will downgrade the importance of our technique. The results here are for an explorable discussion.

## M. Conclusion and Discussions

In this paper, we present AmbiSuR to explore an intrinsic solution for the photometric ambiguity-robust surface reconstruction. Building upon Gaussian Splatting, our work uncovers two built-in primitive-wise representational ambiguities. Then, investigating the wrongly reconstructed ambiguities from unclear supervisions, we reveal an intrinsic potential for ambiguity self-indication from the Spherical Harmonics. With these findings, we design a photometric disambiguation and an ambiguity indication module, constraining ill-posed geometry solutions for a definite surface formation, and unleashing the self-indication potential to identify and direct underconstrained regions to recover correct geometry.

Despite the outstanding performance achieved, so far some limitations still exist. First, due to the lacking support of complex ray propagation in the prevailing rasterization-based 3DGS, scenes with complex lighting environments with specular surfaces still have much room for improvement. Although our Spherical Harmonics Ambiguity Indicator is effective in identifying the ambiguous regions, the underlying photometric calculations are still incorrect in the primary-ray-only rasterization model. Efficiently incorporating ray tracing into the pipeline would be interesting in the future.

Secondly, as in most of the vision-based surface reconstruction methods, identifying transparent surfaces is still a difficult problem. Existing methods are mostly built upon precise light transport calculation (Sun et al., 2024; Huang et al., 2025), which may face challenges when the environment is not fully captured, as in cases of our photometric ambiguity. In the future, the rich priors from learning-based methods may help to solve this long-standing visual problem.

