# OpenReview forum: "Revisiting Photometric Ambiguity for Accurate Gaussian-Splatting Surface Reconstruction"
_ICML.cc/2026/Conference — ICML 2026 regular_

### Official Review · Reviewer_D6fy · 2026-03-06

**Soundness:** 3
**Presentation:** 3
**Significance:** 3
**Originality:** 3
**Overall Recommendation:** 5
**Confidence:** 3

**Summary:**

The paper discusses problems with photometric supervision in the context of surface reconstruction using Gaussian Splatting methods. Authors point at two areas of Gaussian Splatting methods that may lead to ambiguities, which in turn result in suboptimal surface reconstruction. Those are (1) long tails of Gaussian distribution for opacity modeling, (2) lack of mechanism enforcing color consistency for accumulated primitives. They also acknowledge a long standing problem of photometric ambiguity that may lead to misleading supervision. Authors propose solutions for each of this problem, (1) primitive truncation during splatting in order to limit the area of influence, (2) color regularization on primitives intersected by the same ray, (3) indication of ambiguous areas and corresponding primitives to apply additional supervision to them. Supervision is provided by depth estimation models. Using those authors claim to obtain state-of-the-art results on a comprehensive suite of datasets in surface reconstruction task, without major loss in novel-view synthesis task.

**Compliance With Llm Reviewing Policy:**

Affirmed.

**Final Justification:**

After rebuttal I believe this work meets the bar for acceptance and my final score is "accept"

**Key Questions For Authors:**

1. Could the authors clarify why high-order SH magnitude is a reliable indicator of photometric ambiguity rather than simply view-dependent appearance like specular effects?

2. how robust the amorphous mask is, did the authors analyze, how well amorphous mask can detect ill-constrained regions?

3. Could the authors comment on the failure cases of the proposed method on more challenging scenes? Since the paper specifically targets photometric ambiguity, while many existing methods already perform reasonably well on standard datasets, it would be interesting to understand whether the proposed ambiguity-aware constraints enable better reconstruction in more difficult scenarios, such as scenes with strong specularities, highly non-Lambertian materials or other scenarios authors find interesting (this is an open question). Additionally, do the authors observe particular cases where the method still struggles despite the ambiguity handling mechanisms?

**Limitations:**

yes, but Discussion section is at the end of the supplementary material

**Strengths And Weaknesses:**

Soundness

The submission is technically sound. Authors provide sufficient intuition and explanation of their statements in the main part with additional analysis in appendix. Conducted experiments are comprehensive and well structured. Ablation study provides convincing quantitative results, however additional qualitative examples  for all ablated variants added to appendix would be beneficial. Authors are honest with regard to the strengths and weaknesses of their work. But it would be beneficial to clearly state limitations in the main part of the work. It should be clearly emphasized what improvements require additional data obtained from other methods/models.

Presentation

The submission is well written and properly positions itself in the context of previous and concurrent literature. The work is well structured for the most part, however there are few things that should be addressed. It remains unclear whether authors build upon PGSR or just use their codebase for ease of comparison. Preliminaries suggest that the work uses standard 3DGS formulation, at the same time learning objective possess Lgeo factor that is not present in 3DGS and is not introduced in the submission itself. Explicitly stating on which method the work builds on its implementation and exactly what elements of other works are being used. The role of Amorphous Local Regularizer is not obvious at the first read. It would be beneficial to clearly state its purpose before introducing and how it addresses the photometric ambiguity problem.

Significance.

The work addresses the very important problem of limitations of the photometric supervision in the context of mesh reconstruction using 3DGS methods. I believe it is important for 3DGS methods.

Originality

The submission draws on several classic reconstruction works and provides a deeper analysis of known ideas in the context of 3D Gaussian Splatting (3DGS). It introduces a set of constraints and regularizations whose combined formulation has not been presented in prior work, which constitutes the main aspect of its originality. I primarily view the contribution as a strong engineering effort that, in my opinion, meets the bar for acceptance. At the same time, the method offers useful insights for the community and deepens the understanding of photometric supervision in Gaussian-splatting-based reconstruction.

---

> ### Author Rebuttal · Authors · 2026-03-30
>
> ### **Responses to Reviewer D6fy**
>
> We sincerely thank Reviewer D6fy for the time and effort in the constructive and encouraging review. Here we address the specific questions from the reviewer individually:
>
> (Please refer to https://github.com/AmbiSuR/ambisur-rebuttal for the response Figures Rx.)
>
> ---
>
> > **Q1: Explanation of implementation details.**
>
> Thanks for the constructive feedback! We apologize for any lack of clarity in our presentation caused by text simplification. In Eq. (15), we formulated the overall optimization objective in an architecture-agnostic form, where $L_{photo}$ represents the overall training loss (prior-irrelevant) of the used GS backbone, and $L_{geo}$ is our defined variants-related term, as described after Eq. (15). In the experiments, our method is built upon PGSR to get a desired starting point, with all ablation studies kept aligned with or beyond this basis (e.g., Table 4.A, 5.A, and all the others). The training losses of PGSR correspond to $L_{photo}$, and the whole framework still follows the general 3DGS formulation as in preliminaries.  In the revised version, we'll refine the discussion of limitations, external cues, and add qualitative examples for all variants as suggested.
>
> ---
>
> > **Q2: Presentation of the role of Amorphous Local Regularizer.**
>
> Thanks for the thoughtful suggestion. The Amorphous Local Regularizer aims to precisely correct erroneous geometry by selectively applying external geometry priors only to the amorphously distributed primitives, which are identified as ambiguous by SH indicator. This targeted approach compensates for missing or misleading photometric constraints in those specific regions without degrading the surrounding, well-constrained geometry. We will emphasize the effect in the revision.
>
> ---
>
> > **Q3: Explanation of high-order SH magnitude as a reliable indicator.**
>
> Thanks for the feedback. Theoretically, large high-order SH magnitude (Eq. (9)) signifies strong view-dependence. During optimization, this can be divided into two situations: (1) The appearance itself actually changes significantly, which corresponds to ambiguous solutions by factors like reflections or lighting variations during capture. (2) The view-dependence does not exist in the given supervision, but because incorrect geometry was reconstructed, the SH is used to mimic the appearance based on that wrong geometry. Experimentally, as illustrated in Figure 3, not only are the specular regions captured but also largely the second situation, of which both our indicator can effectively reflect for photometric ambiguity.
>
> ---
>
> > **Q4: Performance of amorphous mask in detecting ill-constrained regions.**
>
> Thanks for the thorough consideration. In the paper, we analyzed the robustness and performance of amorphous mask by ablating the individual component (Table 5.E), percentile (Table 9), and visualized its effects in Figure 2 and 3. To quantify the detected ill-constrained regions that lead to low reconstruction accuracy, we conduct an exploratory study here by measuring the depth error in different regions. However, to expose the inaccuracy and avoid underfitting, we'd like to note that this can only be measured where regularization is not applied and only when the optimization is finished (rather than the desired intermediate). Under such non-ideal cases, the consistently concentrated error detection with low area proportion shows the strong robustness and performance of the amorphous mask.
>
> *Table R3.1. (exploratory quantitative study) Masked depth error versus the average. High robustness performance is shown in the non-ideal non-regularization use cases at 30,000th iteration.*
>
> | | Barn | Caterpillar  | Courthouse | Ignatius | Meetingroom  | Truck |
> | --- | --- | --- | --- | --- | --- | --- |
> | L1 Error In Mask | **+ 39.28%** | **+ 46.87%** | **+ 14.28%** | **+ 29.63%** | **+ 15.06%** | **+ 96.29%** |
> | L1 Error Out of Mask | - 7.14% | - 3.12% | - 0.02% | - 0.00% | - 5.47% | - 3.70% |
> | Mask Area Ratio | 13.3% | 9.7% | 20.9% | 3.2% | 27.7% | 6.3% |
>
> ---
>
> > **Q5: Discussion on performance in special cases like transparent and refractive objects.**
>
> Thanks for the constructive suggestions. As briefly discussed in appendix H, we fully agree that strong specularities and highly non-Lambertian materials that include multi-bounce light transport are still of high challenge to handle. While our method is currently outstanding in reconstructing specular surfaces (Figure 4 and Figure R1), the rendering may drop in these areas, as we analyzed in R.zeGy Q1. Another difficult case is the transparent objects like the glass in Figure R1. Especially with heavy refractions, we observe our techniques often encounter failures in this long-standing problem. Although a large improvement is brought by our techniques, a fundamental solution may still lie in explicitly modeling complex light transports. We'll supplement the discussions and provide more samples in the revision.

---

> > ### Author Rebuttal · Reviewer_D6fy · 2026-04-03
> >
> > I thank the authors for the rebuttal answers, I believe the paper meets the bar for acceptance. My final score will be accept (5).

---

> > > ### Author Response · Authors · 2026-04-03
> > >
> > > Dear Reviewer D6fy,
> > >
> > > Thank you for your feedback on our responses! We are deeply grateful for your consistent support of our work, and we sincerely appreciate your recognition with the raised score. Your provided constructive suggestions are invaluable in helping us refine and strengthen our work. These will be carefully integrated in the revised version. Thanks once again for your valuable time and efforts throughout the review process!
> > >
> > > Best regards,
> > >
> > > The authors

---

### Official Review · Reviewer_zeGy · 2026-03-11

**Soundness:** 3
**Presentation:** 3
**Significance:** 3
**Originality:** 3
**Overall Recommendation:** 4
**Confidence:** 4

**Summary:**

This paper introduces AmbiSuR, a framework for 3D Gaussian Splatting (3DGS) designed to improve surface reconstruction by mitigating photometric ambiguities. The authors identify two primary sources of error: primitive edge ambiguity, where low-opacity Gaussian tails cause geometric artifacts, and photometric blending ambiguity, where pixel-wise color mixing leads to under-determined geometry. To address these, the framework proposes:
Gaussian Primitive Truncation: A method to exclude edge regions of Gaussians during opacity calculation based on standard deviation.
Ray-Color Consistency: A constraint that enforces similar optical properties for primitives along a single ray.
SH Ambiguity Indicator: A module utilizing high-degree Spherical Harmonics (SH) coefficients to identify regions with inconsistent photometric constraints for targeted regularization.

**Compliance With Llm Reviewing Policy:**

Affirmed.

**Key Questions For Authors:**

(1) Performance in Surface Reconstruction: In Table 3, AmbiSuR is compared against other Surface Recon. methods, yet its metrics are consistently lower. Why does the proposed method underperform relative to other reconstruction baselines in this specific group?

(2) Indicator Robustness: How does the SH indicator perform in sparse-view scenarios where high-degree coefficients might be prone to overfitting rather than indicating true photometric ambiguity?

**Limitations:**

yes

**Strengths And Weaknesses:**

Strengths
(1) Analytical Soundness: The authors provide a mathematical analysis in Appendix A demonstrating how gradient accumulation in the Gaussian core dominates edge feedback, providing a clear rationale for the truncation strategy.
(2) Computational Efficiency: The use of SH coefficients as a dynamic indicator is a "free-lunch" approach that leverages existing model parameters without requiring external pre-trained geometry models for region identification.
(3) Empirical Performance: The method achieves lower Chamfer distances on the DTU dataset compared to voxel-based methods like GeoSVR and 3DGS-based baselines like PGSR. It also shows improvements in F1-scores on the Tanks and Temples benchmark.

Weaknesses
(1) Novel View Synthesis (NVS) Trade-off: Quantitative results in Table 3 indicate that while surface accuracy improves, the PSNR for outdoor scenes in the Mip-NeRF 360 dataset is slightly lower than some baselines. This suggests that stricter geometric constraints may limit the fitting of complex environmental appearances.
(2) Potential Representation Limitations: The truncation of Gaussian edges, while clarifying surface definition, might reduce the model's ability to represent semi-transparent or extremely fine structures like distant foliage or atmospheric effects.

---

> ### Author Rebuttal · Authors · 2026-03-30
>
> ### **Responses to Reviewer zeGy**
>
> We sincerely thank Reviewer zeGy for recognizing our work and providing valuable comments. We noticed that the main questions from the reviewer regard the analysis of novel view synthesis, and SH ambiguity indicator under sparse views. Here, we address the questions individually:
>
> (Please refer to https://github.com/AmbiSuR/ambisur-rebuttal for the response Figures Rx.)
>
> ------
>
> > **Q1: Analysis on the novel view synthesis performance.**
>
> Thank you for the thorough consideration. In Table 3, AmbiSuR gets top-3 in 5 of the total 6 metrics, which exhibits the competitive performance of our method in rendering. Due to different implementations, the performance in novel view rendering (NVS) of different surface reconstruction approaches actually starts from different original baselines rather than vanilla 3DGS (e.g., GOF adopts Mip-Splatting, PGSR integrates AbsGS, GeoSVR is built upon SVRaster), and therefore leads to different basic NVS performance initially.
>
> To better verify the novel view synthesis performance of our method, besides the official scores in Table 3, we additionally add a comparison here regarding our method and the reproduced PGSR codebase, where the environmental uncertainties are better controlled. While our geometric quality achieves higher, the below results show that the complex outdoor rendering actually gets improved by our approach (which can also be observed in Q2), and the indoor rendering is at a similar level to the previous.
>
> | Outdoor // Indoor          | PSNR  | SSIM  | LPIPS / | / PSNR  | SSIM  | LPIPS |
> | -------------------------- | ----- | ----- | ------- | ------- | ----- | ----- |
> | Reproduced PGSR (Codebase) | 24.68 | 0.749 | 0.206 / | / 30.09 | 0.929 | 0.159 |
> | Ours                       | 24.79 | 0.752 | 0.202 / | / 30.06 | 0.928 | 0.159 |
>
> Similar to the analysis in previous works like GeoSVR, we find that the slight rendering drops in Mip-NeRF 360 indoor scenes are majorly regarding the inherent conflict in the primary-ray-only 3DGS between heavy reflection and accurate geometry. While obtaining more accurate geometry, its reappearance of reflections may be worse than in cases of over-reconstruction with structural errors. Here we list two typical scenes below, where our method consistently obtains more accurate geometry, but the rendering differs.
>
> | Scene-Method (Characteristic)       | PSNR      | SSIM      | LPIPS     |
> | ----------------------------------- | --------- | --------- | --------- |
> | Courter-PGSR (higher reflection)    | **28.32** | 0.913     | **0.173** |
> | Courter-AmbiSuR (higher reflection) | 28.26     | **0.913** | 0.174     |
> | Room-PGSR (lower reflection)        | 30.19     | 0.927     | 0.180     |
> | Room-AmbiSuR (lower reflection)     | **30.21** | **0.928** | **0.177** |
>
> We'll add these analyses in the revised version.
>
> ------
>
> > **Q2: Reconstruction capability for delicate and semi-transparent structures.**
>
> Thanks for your insightful feedback. Gaussian Primitive Truncation would not degrade the capability of reconstructing delicate and semi-transparent structures. As analyzed in Table 8 in the appendix, this technique exhibits an obvious effect by only needing to conduct truncation where the Gaussian weight has already dropped to a much lower level than the core (~0.1-0.01), which ensures the representation capability is maintained as the initials. And in Gaussian Splatting, the transparency of reconstruction is mainly controlled by the individual opacity property for each entire primitive, therefore the truncation will not affect this part. Here we provide the corresponding qualitative results in Figure R3 and R1, which demonstrate our strong or even better reconstruction regarding the distant foliage, atmospheric effects, and semi-transparent glasses. We'll add the results in the revision.
>
> ------
>
> > **Q3: Photometric ambiguity indication under sparse-view scenarios.**
>
> Thanks for the constructive question. The Dual-End Indication in the proposed SH Ambiguity Indicator takes primitives with a large norm of high-degree SH coefficients into consideration, which includes cases of SH overfitting like in sparse-view scenarios. Similar to the shown samples in Figure 3 in the paper, the upper indicator can effectively reflect the inaccurate geometries that using overfitted SH to mimic the photometric appearance, which also works in the mentioned sparse-view cases. In the additional Figure R4, we test the situation using 9 sparse views and compare the indication to the dense-view situation. Besides the observed ambiguity varies due to the changed number of views, the results show that the upper indicator shows high consistency to the situation with dense views. This demonstrates the effectiveness of our method. We'll add this discussion into the revision.

---

> > ### Author Rebuttal · Reviewer_zeGy · 2026-04-06
> >
> > The concerns raised in my review have been clarified. I have no further concerns and thus I keep my positive score on it.

---

> > > ### Author Response · Authors · 2026-04-06
> > >
> > > Dear Reviewer zeGy,
> > >
> > > Thank you for your time in continuously engaging with our responses! It's encouraging to hear that all the concerns have been clarified and no further concerns remain.  Your continued positive evaluation of our work is of great importance to us, and the insightful questions have led us to strengthen our manuscript. Thank you once again for your effort and supportive feedback!
> > >
> > > Best regards,
> > >
> > > The authors

---

### Official Review · Reviewer_BJt5 · 2026-03-12

**Soundness:** 3
**Presentation:** 3
**Significance:** 3
**Originality:** 3
**Overall Recommendation:** 4
**Confidence:** 4

**Summary:**

This paper studies the photometric ambiguity problem in 3D Gaussian Splatting (3DGS) for surface reconstruction. The authors propose AmbiSuR, a framework that introduces several representation-level constraints to mitigate geometry ambiguity arising from photometric supervision. Specifically, the method proposes Gaussian Primitive Truncation to limit geometric over-expansion caused by edge gradients, and Ray-Color Consistency to penalize inconsistent color blending along a ray. Additionally, high-degree Spherical Harmonics (SH) coefficients are used as a signal to detect ambiguous primitives, enabling targeted local regularization guided by external depth priors. Experiments on DTU and Tanks & Temples demonstrate improved surface reconstruction metrics while maintaining competitive novel view synthesis performance.

**Compliance With Llm Reviewing Policy:**

Affirmed.

**Final Justification:**

After the rebuttal phase, my initial concerns are resolved. I believe this work meets the bar, and my final score stands at Weak Accept.

**Key Questions For Authors:**

1. **Transparent and refractive objects:** How does the Ray-Color Consistency loss behave in scenes containing transparent or refractive surfaces? Does the SH-based ambiguity detection mitigate this issue, or does the model degrade in such scenarios?

2. **Temporal stability of primitive selection:** Do primitives frequently oscillate in and out of the ambiguous set due to the percentile-based threshold? If so, does this affect training stability?

3. **Contribution of depth priors:** Can the authors provide an ablation where:
    * depth priors are removed entirely, or
    * depth priors are applied globally without the proposed ambiguity detection?

    This would help isolate the contribution of the proposed mechanism.

4. **Runtime overhead:** What is the training and inference overhead introduced by the Ray-Color Consistency computation compared to vanilla 3DGS?

**Limitations:**

The authors acknowledge that the method may struggle with transparent or refractive materials, which violate the assumptions of the Ray-Color Consistency term. This limitation could be discussed more prominently in the main paper to clarify the applicability scope.

**Strengths And Weaknesses:**

##  Strengths

**1. Clear motivation for addressing photometric ambiguity**

The paper highlights an important limitation of photometric supervision in 3D Gaussian Splatting, where multiple primitives along a ray can produce geometrically ambiguous solutions. The analysis of gradients around Gaussian edges (Appendix A) provides useful intuition for why primitives tend to expand along strong image gradients. This analysis motivates the proposed Gaussian Primitive Truncation mechanism.

**2. SH-based ambiguity detection for localized regularization**

The method uses high-degree spherical harmonic (SH) coefficients as an indicator of photometric ambiguity. Instead of applying global geometric constraints, the approach identifies potentially ambiguous primitives and applies depth regularization locally. This design helps avoid excessive smoothing while still stabilizing geometry.

**3. Comprehensive empirical evaluation**

The experimental section includes comparisons with several recent baselines as well as ablation studies analyzing the proposed components. The paper also evaluates the method with different depth foundation models, suggesting that the performance improvements are not tied to a specific prior.

---

## Weaknesses

**1. Limited novelty relative to recent 3DGS methods**

Although the individual components are well motivated, the overall framework mainly combines several ideas that have appeared in related work, including geometry regularization, ray-based color constraints, and the use of external depth priors. As a result, the methodological novelty may be perceived as incremental. The paper could better clarify how AmbiSuR differs conceptually from existing geometry-regularized splatting approaches.

**2. Assumption behind Ray-Color Consistency**

The Ray-Color Consistency term assumes that primitives contributing to the same ray should have similar colors to represent a consistent surface. While this assumption is reasonable for opaque surfaces, it may not hold in cases involving transparency, refraction, or strong view-dependent effects. The paper would benefit from a clearer discussion of these limitations.

**3. Stability of percentile-based primitive selection**

Ambiguous primitives are selected based on a percentile threshold of SH coefficients. Since Gaussians are continuously split and optimized during training, the SH distribution may change over time. It is unclear whether this could lead to unstable primitive selection or oscillations during training.

**4. Role of external depth priors**

The method relies on monocular depth priors to guide geometry correction in ambiguous regions. Although multiple depth models are tested, it remains unclear how much of the improvement comes from the proposed mechanism versus the strength of the depth prior itself.

**5. Runtime overhead**

The Ray-Color Consistency loss requires computing color variance along rays during training. The paper does not clearly report the additional training cost compared to the vanilla 3DGS pipeline.

---

> ### Author Rebuttal · Authors · 2026-03-30
>
> ### **Responses to Reviewer BJt5**
>
> We sincerely thank Reviewer BJt5 for the positive recognition and insightful feedback on our work. Here we address the questions from the reviewer individually:
>
> (Please refer to https://github.com/AmbiSuR/ambisur-rebuttal for the response Figures Rx.)
>
> ----
>
> > **Q1: Contribution beyond geometry-regularized approaches.**
>
> Thanks for the thoughtful feedback. Different from previous geometry-oriented approaches, our core conceptual contribution is to revisit and reveal the fundamental role of photometric ambiguity in high-accuracy surface reconstruction, delivering a new insight to the community. Specifically, we identify photometric ambiguity as a primary bottleneck that limits reconstruction quality, and analyze its impact from the perspective of the underlying representation and optimization. Based on this insight, AmbiSuR is built to systematically address such ambiguities, leading to more reliable surface recovery in challenging, under-explored regions where previous methods usually fail. We will emphasize this more clearly in the revised version.
>
> ---
>
> > **Q2: Effect of two techniques for transparent surfaces.**
>
> > - How does the Ray-Color Consistency behave in transparent surfaces?
>
> Thanks for your thoughtful consideration. As we discussed in Appendix H, nowadays identifying transparent surfaces, especially with heavy refraction, is one of the most difficult problems that can hardly be resolved without explicit multi-bounce light transport calculation and known environment. Facing this hard problem, Ray-Color Consistency actually provides benefits, relieving the incorrect over-reconstruction from overfitting. To show this in-the-wild performance, here we additionally evaluate our method on the representative EnvGS dataset, which contains real-world transparent surfaces. As shown in Figure R1, Ray-Color Consistency effectively helps reconstruct transparent surfaces with no additional degradation.
>
> > - Can the SH Ambiguity Indicator mitigate the issue?
>
> As shown in Figure R1 above, SH Ambiguity Indicator shows strong effect in this case, especially for surface with complex optical properties, significantly improving the geometry in the wild beyond the naive solutions. These show the effectiveness of our proposals.
>
> Nevertheless, as in the discussion, we acknowledged this is an opening and challenging issue that our method currently cannot well solve, especially for refractive surfaces, which is a valuable future direction. We'll add the discussions in the revised version.
>
> ---
>
> > **Q3: Stability of percentile-based primitive selection.**
>
> Thanks for your insightful comment. In our test, the percentile-based primitive selection keeps high stability during the optimization. While primitives vary and SH distribution shifts, our indicator in practice focuses on the ambiguities from supervision, which keep stable over time, as visualized in Figure R2. Here we additionally conduct ablations on the update intervals below, which show that while selection sets evolve frequently, this dynamic process remains stable and benefits reconstruction accuracy. Combined with Table 9 in the paper, the percentile-based primitive selection exhibits high stability in the training process. We'll add the discussion in the revision.
>
> | Update per Steps | Chamfer / d2s / s2d ↓ |
> | --- | --- |
> | 1 | **0.461** / 0.419 / 0.504 |
> | 10 | **0.461** / 0.417 / 0.505 |
> | 30 | 0.462 / 0.418 / 0.506 |
> | Only After Densification | 0.466 / 0.428 / 0.504 |
> | Vanilla (for Reference) | 0.477 / 0.436 / 0.519 |
>
> ---
>
> > **Q4: Is improvements can be gained from vanilla depth priors?**
>
> Thanks for the constructive suggestion. In Table 4 and 5 in the paper, we ablated the situations where depth priors are applied globally w/o the proposed indicator by E and F in Table 4, A and G in Table 5. Here we supplement the results where depth priors are removed entirely and cases with DA3 metric depth as below. The study shows that our contribution can be well isolated and verified with extensive improvements exceeding the priors' quality. We'll add the results in the revision.
> | TnT | Precision | Recall | F1-Score |
> | --- | --- | --- | --- |
> | Depth Priors Removed Entirely | 0.528 | 0.565 | 0.542 |
> | (Mono) Globally Applied w/o Indicator | 0.533 | 0.592 | 0.557 |
> | (Mono) with Proposed Indicator | **0.568** | **0.594** | **0.576** |
> | (Metric) Globally Applied w/o Indicator | 0.569 | 0.592 | 0.577 |
> | (Metric) with Proposed Indicator | **0.579** | **0.608** | **0.589** |
>
> ---
>
> > **Q5: Overhead of Ray-Color Consistency.**
>
> Thanks for the constructive feedback. Ray-Color Consistency is applied efficiently with marginal overhead. This CUDA-implemented loss adds negligible overhead as required states can be fetched from the rasterizer. Moreover, by reducing redundant primitives, it actually shortens training by ~7 mins (DTU) and ~4 mins (TnT). This technique is not involved in inference. We'll add the analysis in the revision.

---

> > ### Author Rebuttal · Reviewer_BJt5 · 2026-04-03
> >
> > I have read the authors' rebuttal and appreciate the extra experiments. It addresses my initial concerns well.
> > Overall, I am maintaining my current score.

---

> > > ### Author Response · Authors · 2026-04-03
> > >
> > > Dear Reviewer BJt5,
> > >
> > > Thanks for your prompt review of our responses! We are delighted to see the remaining concerns have now been resolved. We are truly grateful for your favorable evaluation from the beginning, and we really appreciate the thoughtful feedback and raised insightful discussions that help us further strengthen the manuscript. Once again, thank you for your invaluable efforts in reviewing our paper!
> > >
> > > Best regards,
> > >
> > > The authors

---

### Decision · Program_Chairs · 2026-04-30

**Decision:**

Accept (regular)

**Comment:**

The work presents an approach for improving surface reconstruction accuracy using 3D Gaussian Splatting, and the reviewers find the method technically sound with promising empirical performance. While one review is missing, the three available reviews and the discussion provide sufficient evidence to support a positive assessment of this paper.

I do not observe any critical issues that would preclude acceptance. Overall, the paper offers useful insights and contributions that are likely to benefit the community.